# TAG: Tangential Amplifying Guidance for Hallucination-Resistant Sampling

Hyunmin Cho [1] [*]  Donghoon Ahn [2] [*]  Susung Hong [3] [*]  Jee Eun Kim [1]
Seungryong Kim [4] [†]  Kyong Hwan Jin [1] [†]

## Abstract

Diffusion models achieve state-of-the-art image generation but often produce semantic inconsistencies, or *hallucinations*. Existing inference-time guidance methods rely on external signals or architectural modifications, adding computational overhead. We propose **T**angential **A**mplifying **G**uidance (**TAG**), a training-free, architecture-agnostic, plug-and-play guidance method that operates purely on trajectory signals. TAG uses an intermediate sample as a projection basis and amplifies the tangential components of the estimated score to correct the sampling trajectory. A first-order Taylor analysis shows that this steers the state toward higher-probability regions of the data manifold, reducing inconsistencies and improving fidelity while adding negligible overhead to existing samplers. Code is available at our Project Page 🜛.

## 1. Introduction

Hallucination in diffusion models refers to the phenomenon of generating samples that violate the data distribution or contradict the conditioning, thereby failing to produce meaningful outputs. For example, it often manifests as mixed-up objects (Okawa et al., 2023; Oriyad et al., 2025) or anatomically implausible structures (e.g., extra-fingered hands). Recent evidence suggests that the primary source of such errors lies in a failure mode known as mode interpolation. During sampling, trajectories may traverse low-density valleys between distinct modes of the data distribution, causing attribute mismatches and structural inconsistencies (Aithal et al., 2024).

A widely adopted remedy involves inference-time guidance

---
[*]Equal contribution [1]Korea University [2]University of California, Berkeley [3]University of Washington [4]KAIST AI . Correspondence to: Seungryong Kim <seungryong.kim@kaist.ac.kr>, Kyong Hwan Jin <kyong_jin@korea.ac.kr>.

*Proceedings of the 43rd International Conference on Machine Learning*, Seoul, South Korea. PMLR 306, 2026. Copyright 2026 by the author(s).

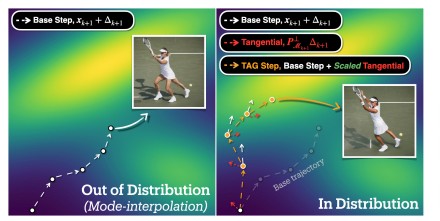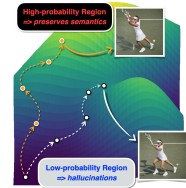

*(a)* No Guidance  *(b)* TAG Update

*Figure 1.* **Conceptual visualization of Tangential Amplifying Guidance (TAG)** from a mode-interpolation perspective (Aithal et al., 2024). Unlike **(a)** no guidance case, **(b)** TAG decomposes the base increment $\Delta_{k+1}$ on the latent sphere into parallel $\boldsymbol{P}_{\mathcal{M}_{k+1}}\Delta_{k+1}$ and orthogonal (i.e., tangential) $\boldsymbol{P}^{\perp}_{\mathcal{M}_{k+1}}\Delta_{k+1}$ components (Equation (7)). By preserving the parallel component while adding a *scaled* tangential component, TAG isolates the data-relevant part of the update (Section 3) and can more effectively navigate the data manifold, leading to samples that contain more semantic structure. We make this precise by proving that amplifying the tangential component has the effect of guiding the trajectories toward regions of higher model density while mitigating off-manifold drift (Section 4 and Equation (18)).

strategies, such as classifier-free guidance (CFG) (Ho & Salimans, 2021) and their variants (Hong et al., 2023; Ahn et al., 2024; Hong, 2024; Rajabi et al., 2025; Kwon et al., 2025; Sadat et al., 2025; Dinh et al., 2025; Kim & Sim, 2025). Under the assumption that deviating from low-probability regions enhances sample quality, most of these methods employ *residual scaling*, using the difference between the conditional and unconditional branches to guide the generation process away from the unconditional model's outputs. While effective, such guidance is largely *geometry-unaware*: it applies a *single scalar magnification* to the cond–uncond residual, without accounting for the *local directional structure* of the data distribution at each noise level, which can inadvertently distort the denoising trajectory.

Motivated by this, we instead adopt a score-level view of guidance grounded in Tweedie's identity (Tweedie, 1984), which relates the score to the posterior mean under Gaussian corruption. This link motivates a decomposition of the model update based on its *intrinsic geometry*: a drift component that advances the radius along the prescribed noise schedule (i.e., noise level), and a tangential component that moves along the *data-manifold*, approximately preserving the overall *radius* while refining the sample's structure and semantics. We observe that the tangential component carries rich structural information (Figure 2), and amplifying it

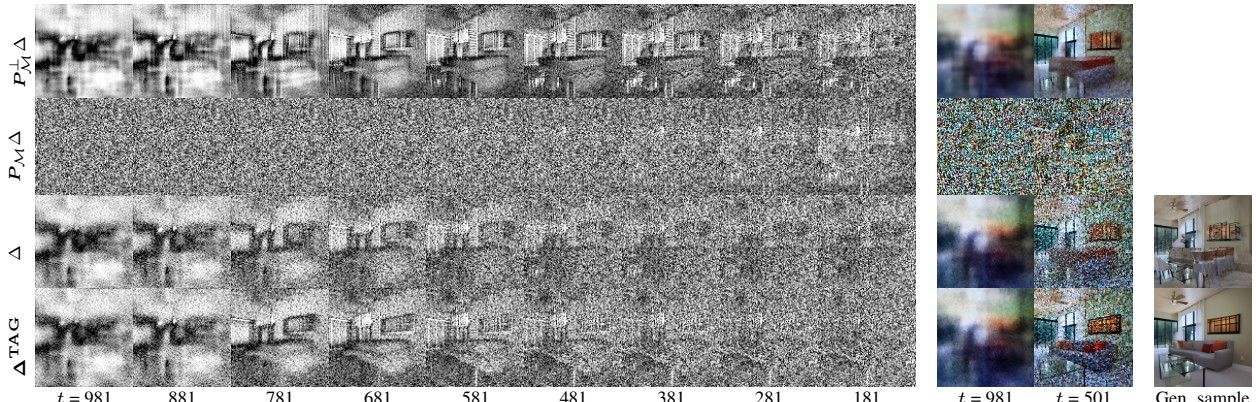

*Figure 2.* **Amplifying the tangential component enhances semantic content by isolating it from noise.** This figure illustrates the decomposition of the update step $\Delta$ into *normal* and *tangential* components. Subtracting the unstructured, noisy normal component $\boldsymbol{P}_{\mathcal{M}}\Delta$ from the original update acts as a *denoising operation*, revealing the tangential component $\boldsymbol{P}_{\mathcal{M}}^{\perp}\Delta$, which preserves the *principal semantic structure*. Images decoded from intermediate timesteps ($t = 981, 501$) indicate that semantic information is most salient in the tangential component. Motivated by this observation, our method $\boldsymbol{\Delta}^{\mathbf{TAG}}$ amplifies this semantically rich component, yielding a clearer and more coherent final sample (far right) than that obtained from the unmodified $\Delta$.

reduces out-of-distribution samples (Figure 3).

Drawing upon the principle of *amplifying the tangential component* during inference, we derive **T**angential **A**mplifying **G**uidance (**TAG**), a plug-and-play method that emphasizes the tangential component of the *score update*. TAG steers the sampling trajectory to closely follow the underlying data manifold. TAG integrates seamlessly with standard diffusion backbones—whether conditioned or not—without requiring additional denoising evaluations or retraining.

We can summarize our contributions as follows:

- We establish a concrete link between the score's intrinsic geometry and sample quality, proving that amplifying the tangential components steers sampling trajectories toward the in-distribution manifold.

- We introduce TAG, a theoretically grounded, computationally lightweight, and architecture-agnostic algorithm that realizes this geometric principle in practice.

## 2. Preliminaries

**Score-based Diffusion Model.** Score-based generative models learn a time-indexed score function that approximates the gradient of the log-density of noise-perturbed data,

$$\boldsymbol{s}_\theta(\boldsymbol{x}, t_k) \approx \nabla_{\boldsymbol{x}} \log p(\boldsymbol{x} \mid t_k),$$

$t_k \in \{t_K > \cdots > t_0\}$ denotes the $k$-th discretized timestep, to reverse a gradual noising process for sample generation. This approach provides a continuous-time framework that unifies earlier discrete-time Denoising Diffusion Probabilistic Models (DDPMs) (Sohl-Dickstein et al., 2015; Ho et al., 2020) through the lens of stochastic differential equations (SDEs) (Song et al., 2021b). The core idea involves a forward-time SDE that transforms complex data into a simple prior distribution, given by

$$d\boldsymbol{x} = \mathbf{f}(\boldsymbol{x}, t)dt + g(t)d\mathbf{W}.$$

Generation is then performed by the corresponding reverse-time SDE, which becomes tractable by substituting the unknown true score with the learned model $\boldsymbol{s}_\theta$ (Anderson, 1982). To solve this numerically, we discretize the time horizon over timesteps $t_k \in \{t_K > \cdots > t_0\}$. This score network, typically a noise-conditional U-Net, is trained efficiently via denoising score matching across various noise levels (Vincent, 2011; Song & Ermon, 2019). For sampling, one can use numerical methods such as predictor-corrector schemes to simulate the stochastic reverse SDE, or solve an associated deterministic ordinary differential equation (ODE), known as the probability-flow ODE. This continuous-time framework not only provides a theoretical basis for widely used deterministic samplers like DDIM (Song et al., 2021a), but also has inspired modern refinements, such as preconditioning and parameterization in EDM (Karras et al., 2024b), which further enhance the trade-off between sample quality and efficiency.

**Inference-Time Guidance.** Numerous methods modify the update field during sampling to improve fidelity without retraining. Early CFG-style guidance (Ho & Salimans, 2021) steers samples by rescaling residual signals, and complementary approaches replace external cues with model-internal signals for guidance (Hong et al., 2023; Ahn et al., 2024; Hong, 2024). However, prior analyses show that naïve, geometry-agnostic scaling can reduce diversity or perturb solver dynamics (Dhariwal & Nichol, 2021; Kynkäänniemi et al., 2024). These limitations motivate geometry-aware guidance that asks not only how much to scale, but which directions to emphasize. Representative methods along this line use projections to dampen undesired components (e.g., high-scale saturation or cond–uncond mismatch) (Armand-pour et al., 2023; Sadat et al., 2025; Kwon et al., 2025), or

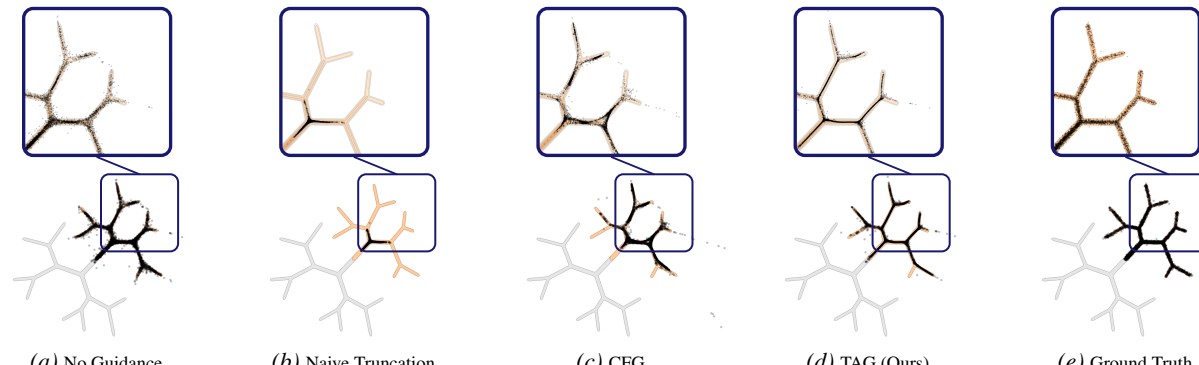

*(a)* No Guidance     *(b)* Naive Truncation     *(c)* CFG     *(d)* TAG (Ours)     *(e)* Ground Truth

*Figure 3.* **Sampling on a 2D branching distribution (Karras et al., 2024a) under different guidance methods.** (a) *No guidance*: probability mass drifts off the data manifold, yielding fragmented branches and OOD (Out of Distribution) points. (b) *Naive truncation*: suppresses some OOD but oversimplifies the geometry, dropping fine branches. (c) *CFG*: reduces boundary violations but also reduces diversity and can still leave OOD strays in our run. (d) **TAG (Ours)**: trajectories are steered toward high-density regions along the branches, suppressing off-manifold outliers while retaining detail. (e) *Ground truth*. Overall, TAG achieves the highest similarity to the GT distribution **without additional #NFEs**, concentrating mass on the correct branches while reducing OOD outliers.

are developed for more specific problem settings such as loss-guided inverse problems or unpaired I2I (Sun et al., 2023; He et al., 2024) (see Appendix E for more detailed discussion). Overall, these strategies integrate cleanly with modern solvers and effectively suppress off-manifold drift, but are often closely tied to particular guidance algebra or task-specific assumptions.

## 3. Motivation and Intuition

Under Gaussian corruption, Tweedie's formula (Tweedie, 1984) links the posterior mean of the clean signal to the noisy observation via the score (i.e., the gradient of the log marginal density):

$$\mathbb{E}[\boldsymbol{x}_0 \mid \boldsymbol{x}_k] = \frac{1}{\sqrt{\bar{\alpha}_k}} \Big( \boldsymbol{x}_k + \overbrace{\sigma_k^2 \nabla_{\boldsymbol{x}} \log p(\boldsymbol{x} \mid t_k)\big|_{\boldsymbol{x}=\boldsymbol{x}_k}}^{\triangleq \text{ Tweedie Increment } \Delta_k^{\text{Tw}}} \Big),$$
(1)

where $\boldsymbol{x}_k$ denotes the noisy observation and $\Delta_k^{\text{Tw}}$ denotes the data-driven correction term. Geometrically, the score field $\nabla_{\boldsymbol{x}} \log p(\boldsymbol{x}|t_k)\big|_{\boldsymbol{x}=\boldsymbol{x}_k}$ points in the direction of steepest increase of the marginal density. Tweedie's formula therefore adjusts $\boldsymbol{x}_k$ in this ascent direction, nudging the state toward higher-probability regions. Therefore, the aim of modeling is to accurately capture these data-driven directions.

To get better intuition for these *data-driven directions*, we appeal to the Gaussian annulus theorem (Blum et al., 2020), which states that an isotropic Gaussian in $\mathbb{R}^d$ concentrates most of its mass near a thin spherical shell. Since each corruption step adds such noise to the data, the corrupted distribution $p(\boldsymbol{x}|t_k)$ likewise places most of its mass near a thin shell. Consequently, we can regard the high-probability region of $p(\boldsymbol{x} \mid t_k)$ as a *noisy data manifold* $\mathcal{M}_k$ embedded near this thin shell. Our goal is therefore to move *along* the high-probability shell of the corrupted distribution, rather than pushing the sample inward or outward in radius, which is already dictated by the diffusion noise schedule.

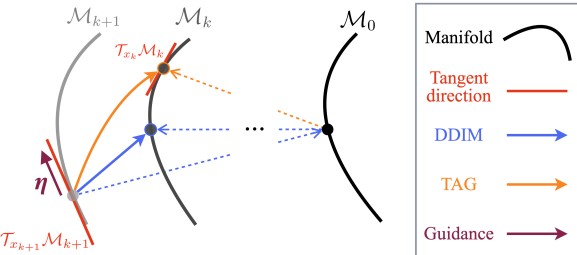

*Figure 4.* **Illustration of Tangential Amplification.** The curves $\mathcal{M}_k$ denote the noisy data manifolds at successive diffusion steps. TAG decomposes $\Delta_{k+1}$ into normal and tangential components with respect to $\mathcal{M}_{k+1}$, and amplifies the tangential component within the local tangent space $\mathcal{T}_{x_{k+1}}\mathcal{M}_{k+1}$ by a factor $\eta$.

To separate the radial (normal-to-shell) component from the within-shell (surface-direction) component, we define:

$$\boldsymbol{P}_{\mathcal{M}_k} = \widehat{\boldsymbol{x}}_k \widehat{\boldsymbol{x}}_k^\top, \quad \boldsymbol{P}_{\mathcal{M}_k}^\perp = I - \boldsymbol{P}_{\mathcal{M}_k}$$
$$\text{where} \quad \widehat{\boldsymbol{x}}_k := \boldsymbol{x}_k / \|\boldsymbol{x}_k\|_2,$$
(2)

where $\boldsymbol{P}_{\mathcal{M}_k}$ projects onto the radial direction (normal to the shell) and $\boldsymbol{P}_{\mathcal{M}_k}^\perp$ onto the within-shell directions (Figure 4).

Guided by this separation, we form the amplified state $\boldsymbol{x}^+$

$$\boldsymbol{x}^+ = \big( \boldsymbol{x}_k + \boldsymbol{P}_{\mathcal{M}_k} \Delta_k^{\text{Tw}} \big) + \eta \, \boldsymbol{P}_{\mathcal{M}_k}^\perp \Delta_k^{\text{Tw}}, \quad \text{with} \quad \eta \geq 1.$$
(3)

By doing so, we preserve the radial first-order term (Equation (22)) while allocating additional gain to the within-shell component of the Tweedie increment, steering the sampling step toward higher-density regions along $\mathcal{M}_k$. In the following section (Section 4.1), we formalize this idea as a *constrained MLE* update that allocates first-order gain to the tangential subspace.

## 4. TAG: Tangential Amplifying Guidance

We introduce Tangential Amplifying Guidance (TAG), which reweights base increments along normal/tangential directions on the latent space.

**Definitions & Algorithm.** We work per sample on $\mathbb{R}^{C \times H \times W} \cong \mathbb{R}^d$ with Euclidean inner product $\langle \cdot, \cdot \rangle$ and norm $\| \cdot \|_2$. Let $\{t_k\}_{k=K}^0$ be *descending* timesteps with $t_K > \cdots > t_0$, and let $\epsilon_\theta$ denote the denoiser. Given $\boldsymbol{x}_{k+1}$ at time $t_{k+1}$, the denoiser predicts

$$\boldsymbol{\varepsilon}_{k+1} \;=\; \epsilon_\theta(\boldsymbol{x}_{k+1}, t_{k+1}).$$

A base solver (e.g., DDIM) then produces a *provisional* state (Karras et al., 2024b)

$$\tilde{\boldsymbol{x}}_k \;=\; a_{k+1}\boldsymbol{x}_{k+1} \;+\; b_{k+1}\boldsymbol{\varepsilon}_{k+1}, \tag{4}$$

where $a_{k+1}, b_{k+1}$ are base solver coefficients. Corresponding base increment at $\boldsymbol{x}_{k+1}$ is defined as

$$\Delta_{k+1} \;:=\; \tilde{\boldsymbol{x}}_k - \boldsymbol{x}_{k+1}. \tag{5}$$

For any $\boldsymbol{x} \in \mathbb{R}^d$, we define the unit vector and orthogonal projectors

$$\boldsymbol{P}_{\mathcal{M}}(\boldsymbol{x}) = \widehat{\boldsymbol{x}}\widehat{\boldsymbol{x}}^\top, \; \boldsymbol{P}_{\mathcal{M}}^\perp(\boldsymbol{x}) = \boldsymbol{I} - \boldsymbol{P}_{\mathcal{M}}(\boldsymbol{x}), \; \text{where } \widehat{\boldsymbol{x}} = \frac{\boldsymbol{x}}{\|\boldsymbol{x}\|_2}. \tag{6}$$

Figure 4 illustrates this tangential–normal decomposition along the sampling trajectory. Given positive scales $\eta \geq 1$, TAG *reweights* the base increment at $\boldsymbol{x}_{k+1}$:

$$\boldsymbol{x}_k \leftarrow \boldsymbol{x}_{k+1} + \boldsymbol{P}_{\mathcal{M}_{k+1}}\Delta_{k+1} + \eta\,\boldsymbol{P}_{\mathcal{M}_{k+1}}^\perp\Delta_{k+1}$$
$$\text{where } \boldsymbol{P}_{\mathcal{M}_{k+1}} = \boldsymbol{P}_{\mathcal{M}}(\boldsymbol{x}_{k+1}), \boldsymbol{P}_{\mathcal{M}_{k+1}}^\perp = \boldsymbol{P}_{\mathcal{M}}^\perp(\boldsymbol{x}_{k+1}). \tag{7}$$

### 4.1. Why does TAG improve Image Quality?

**Log-likelihood maximization.** A foundational goal of training generative models is to maximize the log-likelihood of the data, as formalized by the Maximum Likelihood Estimation (MLE) principle:

$$\max_\theta \sum_i \log p_\theta(\boldsymbol{x}_i). \tag{8}$$

This principle suggests that high-quality samples should concentrate in regions of high probability. To connect this idea to an *update rule*, we relate likelihood increase to movement along the score via a local linearization:

$$\log p_\theta(\boldsymbol{x}) = \log p_\theta(\boldsymbol{x}_0)$$
$$+ (\boldsymbol{x} - \boldsymbol{x}_0)^\top \nabla_{\boldsymbol{x}} \log p_\theta(\boldsymbol{x})\big|_{\boldsymbol{x}=\boldsymbol{x}_0} + \mathcal{O}(\|\cdot\|^2). \tag{9}$$

Diffusion models (Ho et al., 2020; Song et al., 2021b) are designed to predict a score function, $\nabla_{\boldsymbol{x}} \log p(\boldsymbol{x} \mid t_k)\big|_{\boldsymbol{x}=\boldsymbol{x}_k} \approx -\epsilon_\theta(\boldsymbol{x}_k, t_k)/\sigma_k$, which operates on noisy versions of the data. Because diffusion models learn this score field, optimizing the global likelihood (Equation (8)) for a sample $\boldsymbol{x}_0$ during inference is *not directly tractable*. Therefore, we propose to apply the spirit of MLE at each *local step* of the sampling trajectory.

$$\log p(\boldsymbol{x}_k|t_{k+1}) \approx \log p(\boldsymbol{x}_{k+1}|t_{k+1}) + (\boldsymbol{x}_k - \boldsymbol{x}_{k+1})^\top$$
$$\nabla_{\boldsymbol{x}} \log p(\boldsymbol{x} \mid t_{k+1})\big|_{\boldsymbol{x}=\boldsymbol{x}_{k+1}} + \mathcal{O}(\|\cdot\|^2). \tag{10}$$

---

**Algorithm 1** Tangential Amplifying Guidance (**TAG**)

1: **Input:** Denoiser $\epsilon_\theta(\cdot)$, timesteps $\{t_k\}_{k=K}^0$, base solver coefficients $a_{k+1}, b_{k+1}$, TAG scale $\eta \geq 1$
2: Sample $\boldsymbol{x}_K \sim \mathcal{N}(\boldsymbol{0}, \boldsymbol{I})$
3: **for** $k = K-1$ **down to** 0 **do**
4:     $\boldsymbol{\varepsilon}_{k+1} \leftarrow \epsilon_\theta(\boldsymbol{x}_{k+1}, t_{k+1})$
5:     $\tilde{\boldsymbol{x}}_k \leftarrow a_{k+1}\boldsymbol{x}_{k+1} + b_{k+1}\boldsymbol{\varepsilon}_{k+1}$
6:     $\Delta_{k+1} \leftarrow \tilde{\boldsymbol{x}}_k - \boldsymbol{x}_{k+1}$
7:     $\widehat{\boldsymbol{x}}_{k+1} \leftarrow \boldsymbol{x}_{k+1}/\|\boldsymbol{x}_{k+1}\|_2$
8:     $\boldsymbol{P}_{\mathcal{M}_{k+1}} \leftarrow \widehat{\boldsymbol{x}}_{k+1}\widehat{\boldsymbol{x}}_{k+1}^\top$
9:     $\boldsymbol{P}_{\mathcal{M}_{k+1}}^\perp \leftarrow \boldsymbol{I} - \boldsymbol{P}_{\mathcal{M}_{k+1}}$
10:     $\boldsymbol{x}_k \leftarrow \boldsymbol{x}_{k+1} + \boldsymbol{P}_{\mathcal{M}_{k+1}}\Delta_{k+1} + \eta\,(\boldsymbol{P}_{\mathcal{M}_{k+1}}^\perp\Delta_{k+1})$
11: **end for**
12: **Output:** $\boldsymbol{x}_0$

---

The idea of enhancing a pre-trained score function with inference-time guidance has proven effective. For instance, when the score function is well-trained on given training sets, and this leads to a well-trained maximum log-likelihood, we observe that the pre-trained score function can be improved by CFG (Ho & Salimans, 2021), which linearly steers the score toward the conditional target. Inspired by this, our approach provides inference-time guidance on the score function by maximizing the following local log-likelihood term *over admissible one-step updates* (with a step budget $\delta_k$), thereby guiding the sampling trajectory towards high-likelihood regions of the data distribution and reducing off-manifold artifacts (hallucination):

$$\max_{\boldsymbol{x}_k \,:\, \mathcal{C}}(\boldsymbol{x}_k - \boldsymbol{x}_{k+1})^\top \nabla_{\boldsymbol{x}} \log p(\boldsymbol{x} \mid t_{k+1})\big|_{\boldsymbol{x}=\boldsymbol{x}_{k+1}},$$
$$\text{where } \mathcal{C} := \|\boldsymbol{x}_k - \boldsymbol{x}_{k+1}\|_2 \leq \delta_k \tag{11}$$

**Single-step increment decomposition.** For deterministic DDIM/ODE samplers, the *single-step score state decomposition* can be written as

$$\Delta_{k+1} := \tilde{\boldsymbol{x}}_k - \boldsymbol{x}_{k+1} \;=\; \tilde{\alpha}_k\epsilon_\theta(\boldsymbol{x}_{k+1}, t_{k+1}) + \beta_k\boldsymbol{x}_{k+1}, \tag{12}$$

with coefficients

$$\tilde{\alpha}_k := \sigma_k - \frac{\sqrt{\bar{\alpha}_k}}{\sqrt{\bar{\alpha}_{k+1}}}\sigma_{k+1}, \quad \beta_k := \frac{\sqrt{\bar{\alpha}_k}}{\sqrt{\bar{\alpha}_{k+1}}} - 1,$$
$$\text{with} \quad \tilde{\alpha}_k < 0, \; \beta_k > 0, \tag{13}$$

where $\bar{\alpha}$ is the standard diffusion cumulative product term. Using the projection operators, which satisfy

$$\boldsymbol{P}_{\mathcal{M}_{k+1}}^\perp \boldsymbol{x}_{k+1} = 0, \quad \boldsymbol{P}_{\mathcal{M}_{k+1}}\boldsymbol{x}_{k+1} = \boldsymbol{x}_{k+1}, \tag{14}$$

yields the *projection-wise* identities

$$\boldsymbol{P}_{\mathcal{M}_{k+1}}^\perp \Delta_{k+1} = \tilde{\alpha}_k \boldsymbol{P}_{\mathcal{M}_{k+1}}^\perp \epsilon_\theta(\boldsymbol{x}_{k+1}, t_{k+1}),$$
$$\boldsymbol{P}_{\mathcal{M}_{k+1}}\Delta_{k+1} = \tilde{\alpha}_k\boldsymbol{P}_{\mathcal{M}_{k+1}}\epsilon_\theta(\boldsymbol{x}_{k+1}, t_{k+1}) + \beta_k\boldsymbol{x}_{k+1}. \tag{15}$$

Substituting Equation (15) into the Equation (7) gives

$$\boldsymbol{x}_k^{\text{TAG}} = \boldsymbol{x}_{k+1} + \tilde{\alpha}_k\big[\boldsymbol{P}_{\mathcal{M}_{k+1}} + \eta\boldsymbol{P}_{\mathcal{M}_{k+1}}^\perp\big]\epsilon_\theta(\boldsymbol{x}_{k+1}, t_{k+1})$$
$$+ \beta_k\boldsymbol{x}_{k+1}, \quad \text{with} \quad \eta \geq 1. \tag{16}$$

Therefore, the TAG update $\Delta_{k+1}^{\mathrm{TAG}}$ can be expressed in terms of the decomposed components of the original update $\Delta_{k+1}$:

$$\Delta_{k+1}^{\mathrm{TAG}} = \big(\boldsymbol{P}_{\mathcal{M}_{k+1}} + \eta\,\boldsymbol{P}_{\mathcal{M}_{k+1}}^{\perp}\big)\Delta_{k+1}. \qquad (17)$$

In this way, as visualized in Figure 2, semantic information can be isolated from the update vector via the tangential projection, thereby enabling semantics-aware amplification. To quantify its effect on the log-likelihood, assume the log-density is smooth (i.e., $\log p(\cdot|t_{k+1})$ is $C^2$ in a neighborhood of $\boldsymbol{x}_{k+1}$). The *first-order Taylor expansion gain* for a small TAG update $\Delta_{k+1}^{\mathrm{TAG}} \in \mathbb{R}^d$ is

$$G(\eta) := \big(\Delta_{k+1}^{\mathrm{TAG}}\big)^{\top}\nabla_{\boldsymbol{x}}\log p(\boldsymbol{x}\mid t_{k+1})\big|_{\boldsymbol{x}=\boldsymbol{x}_{k+1}}. \qquad (18)$$

Next, we prove that increasing $\eta$ provides a monotonic increase in this first-order gain.

**Theorem 4.1** (Monotonicity of the First-order Taylor Gain). *Assume a deterministic base step with* $\Delta_{k+1} = \tilde{\alpha}_k\epsilon_{\theta}(\boldsymbol{x}_{k+1}, t_{k+1}) + \beta_k\boldsymbol{x}_{k+1}$ *and* $\tilde{\alpha}_k \le 0$. *Let* $\boldsymbol{P}_{\mathcal{M}_{k+1}} \succeq 0$ *and* $\boldsymbol{P}_{\mathcal{M}_{k+1}}^{\perp} \succeq 0$ *be the projectors defined above. For the TAG step* $\Delta_{k+1}^{\mathrm{TAG}} = \boldsymbol{P}_{\mathcal{M}_{k+1}}\Delta_{k+1} + \eta\,\boldsymbol{P}_{\mathcal{M}_{k+1}}^{\perp}\Delta_{k+1}$, *the first-order Taylor gain* $G(\eta) := \big(\Delta_{k+1}^{\mathrm{TAG}}\big)^{\top}\nabla_{\boldsymbol{x}}\log p(\boldsymbol{x}\mid t_{k+1})\big|_{\boldsymbol{x}=\boldsymbol{x}_{k+1}}$ *satisfies*

$$\frac{\partial G(\eta)}{\partial \eta} \approx \frac{-\tilde{\alpha}_k}{\sigma_{k+1}}\big\|\boldsymbol{P}_{\mathcal{M}_{k+1}}^{\perp}\epsilon_{\theta}(\boldsymbol{x}_{k+1}, t_{k+1})\big\|_2^2 \ge 0, \tag{19}$$

*and, in particular,*

$$G^{\mathrm{TAG}} - G^{\mathrm{base}} = \overbrace{-\sigma_{k+1}^{-1}\cdot\big(\tilde{\alpha}_k(\eta - 1)\big)}^{\ge\,\mathbf{0}\ as\ \tilde{\alpha}_k\,\le\,0} \\ \cdot\big\|\boldsymbol{P}_{\mathcal{M}_{k+1}}^{\perp}\epsilon_{\theta}(\boldsymbol{x}_{k+1}, t_{k+1})\big\|_2^2 \ge 0, \tag{20}$$

*Equality holds iff* $\eta = 1$ *or the tangential component of the score is zero. The proof is provided in Appendix Section A.*

**Log-likelihood improvements via TAG.** We cast inference-time guidance as maximizing a log-likelihood gain (Equation (11)). TAG simply reweights the update step by amplifying the component that is orthogonal to the current state while leaving the parallel component unchanged. By Theorem 4.1, increasing the orthogonal weight monotonically raises the first-order Taylor gain, so TAG steers the sampler toward higher-density regions of the data manifold, improving image quality.

**Avoidance of normal amplification.** Amplifying the tangential component monotonically increases the first-order term of a Taylor gain of $\log p(\cdot\mid t_{k+1})$ (Theorem 4.1), which produces samples with fewer hallucinations. However, amplifying the normal component increases radial contraction and leads to over-smoothing (Figure 5). This radial component of the single-step is aligned with the radial

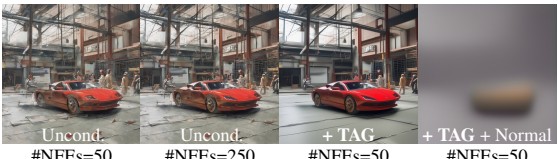

| Uncond. | Uncond. | + TAG | + TAG + Normal |
| #NFEs=50 | #NFEs=250 | #NFEs=50 | #NFEs=50 |

*Figure 5.* **Effectiveness of TAG**. At 50 NFEs, TAG surpasses the sample quality at 250 NFEs from baseline. In contrast, +*Normal* causes severe over-smoothing.

part of Tweedie's correction, which links $\boldsymbol{x}_k$ to the posterior mean $\mathbb{E}[\boldsymbol{x}_0|\boldsymbol{x}_k]$ via the score function (Tweedie, 1984; Song et al., 2021b). Formally, rescaling the normal part by a $\kappa\,(>1)$, the radial first–order change is multiplied by $\kappa$:

$$\langle\widehat{\boldsymbol{x}}_{k+1}, \Delta_{k+1}^{(\kappa)}\rangle = \kappa\,\langle\widehat{\boldsymbol{x}}_{k+1}, \Delta_{k+1}\rangle. \qquad (21)$$

Therefore, a value of $\kappa\,(>1)$ excessively strengthens this contraction under the VP/DDIM schedule, leading to over-smoothing. In contrast, tangential scaling preserves the radial first–order term:

$$\langle\widehat{\boldsymbol{x}}_{k+1}, \Delta_{k+1}^{\mathrm{TAG}}\rangle = \langle\widehat{\boldsymbol{x}}_{k+1}, \Delta_{k+1}\rangle. \qquad (22)$$

Thus, normal amplification breaks one–step calibration and *induces over-smoothing*, whereas tangential boosting improves alignment without disturbing the radial schedule.

### 4.2. TAG for Classifier-Free Guidance

Sections 3 and 4.1 show that the *tangential* component encodes data-relevant directions and is radius-preserving to first-order; so amplifying it improves image quality by steering updates along data-aligned directions. In CFG (Ho & Salimans, 2021), the guided score combines conditional and unconditional branches

$$\widetilde{\varepsilon}_k = \epsilon_{\theta}(\boldsymbol{x}_k, \emptyset) + \omega(\epsilon_{\theta}(\boldsymbol{x}_k, \boldsymbol{c}) - \epsilon_{\theta}(\boldsymbol{x}_k, \emptyset)). \qquad (23)$$

Because these two scores follow distinct trajectories, an incoherence between them can arise, and such an effect can degrade generation quality, an issue recently highlighted by Kwon et al. (2025). Motivated by this established score mismatch, and informed by our core intuition that the tangential field encodes data geometry (Equation (1)), we posit that this incoherence is fundamentally tangential in nature; that is, a persistent *mismatch* exists primarily between the conditional and unconditional tangential components.

**Conditional–unconditional tangent reconciliation.** Let $\boldsymbol{\varepsilon}_u := \epsilon_{\theta}(\boldsymbol{x}_k, \emptyset)$ and $\boldsymbol{\varepsilon}_c := \epsilon_{\theta}(\boldsymbol{x}_k, \boldsymbol{c})$ denote the unconditional/conditional predicted noise, and let $\boldsymbol{g}_k := \boldsymbol{\varepsilon}_c - \boldsymbol{\varepsilon}_u$ be the usual CFG residual. We extract its tangential component via the projector $\boldsymbol{P}_{\mathcal{M}}^{\perp}(\boldsymbol{x}_k)$ and define the *conditional-relative tangent*

$$\boldsymbol{g}_k^{\perp} := \boldsymbol{P}_{\mathcal{M}}^{\perp}(\boldsymbol{x}_k)\boldsymbol{g}_k = \boldsymbol{P}_{\mathcal{M}}^{\perp}(\boldsymbol{x}_k)\big(\boldsymbol{\varepsilon}_c - \boldsymbol{\varepsilon}_u\big). \qquad (24)$$

We then align the conditional prediction with this tangential residual direction by projecting $\boldsymbol{\varepsilon}_c$ onto $\mathrm{span}(\boldsymbol{g}_k^{\perp})$:

$$\boldsymbol{g}_k^{\mathrm{align}} := \frac{\langle\boldsymbol{\varepsilon}_c, \boldsymbol{g}_k^{\perp}\rangle}{\|\boldsymbol{g}_k^{\perp}\|_2^2}\,\boldsymbol{g}_k^{\perp}. \qquad (25)$$

*Table 1.* **Quantitative comparison of *unconditional generation* by** SD-series on COCO 2014.

| Uncond. | FID ↓ | IS ↑ | AES ↑ | CMMD ↓ |
|---|---|---|---|---|
| SD v1.5 | 58.41 | 15.59 | 5.003 | 1.069 |
| **TAG**$_{SD\ v1.5}$ | **46.20** | **16.77** | **5.064** | **0.778** |
| SD v2.1 | 78.54 | 12.52 | 5.299 | 1.395 |
| **TAG**$_{SD\ v2.1}$ | **59.94** | **13.36** | **5.320** | **1.122** |
| SDXL | 119.14 | **9.08** | 5.645 | 2.474 |
| **TAG**$_{SDXL}$ | **90.71** | 8.91 | 5.577 | **2.201** |
| SD3 | 84.26 | 11.53 | 5.261 | 1.671 |
| **TAG**$_{SD3}$ | **79.11** | **11.73** | **5.365** | **1.564** |

*Table 2.* **Quantitative comparison of *conditional generation* by** SD-series on COCO 2014.

| Cond. | FID ↓ | ImageReward ↑ | CLIP ↑ |
|---|---|---|---|
| SD v1.5 | 33.49 | -0.342 | 25.00 |
| **TAG**$_{SD\ v1.5}$ | **26.61** | **-0.339** | **25.09** |
| SD v2.1 | 26.12 | 0.143 | 25.35 |
| **TAG**$_{SD\ v2.1}$ | **21.59** | **0.424** | **26.16** |
| SDXL | 29.28 | 0.274 | 25.41 |
| **TAG**$_{SDXL}$ | **28.53** | **0.292** | **25.49** |
| SD3 | 29.02 | 1.030 | 26.39 |
| **TAG**$_{SD3}$ | **27.54** | **1.043** | **26.56** |

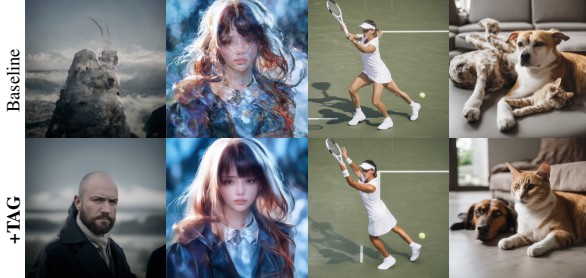

**Unconditional Gen.** (SD3)    **Conditional Gen.** (SDXL)    **Plug-and-Play:** PAG+SDXL    **Plug-and-Play:** SEG+SDXL

*Figure 6.* **Qualitative comparison of TAG.** *Left* four columns demonstrate that TAG enhances the detail and coherence of samples from SDXL/3 under both unconditional and conditional generation. *Right* four columns illustrate its plug-and-play compatibility: TAG can be applied on top of existing methods (e.g., PAG (Ahn et al., 2024), SEG (Hong, 2024)) to further improve their outputs.

Finally, we define TAG on CFG as

$$\tilde{\varepsilon}_k = \varepsilon_u + \omega\,\boldsymbol{g}_k + \eta\,\boldsymbol{g}_k^{\text{align}}, \qquad (26)$$

where $\omega$ is the usual CFG scale and $\eta$ controls the extra tangential emphasis. *Further implementation details are provided in Appendix Section D.*

## 5. Experiments

**General evaluation setup.** We apply TAG at inference time to pretrained backbones, including Stable Diffusion v1.5/v2.1/XL (Rombach et al., 2022; Podell et al., 2024) and Stable Diffusion 3 (Esser et al., 2024), which follows a flow-matching formulation (Lipman et al., 2023; Liu et al., 2023). We also use representative guidance methods, PAG (Ahn et al., 2024) and SEG (Hong, 2024), as comparison baselines and plug-and-play modules. *We further evaluate TAG with SSG (Zhang et al., 2026) in Appendix Section B.1 to assess its applicability across different guidance schemes.*

**General evaluation metrics and protocol.** We report FID (Seitzer, 2020) (distributional similarity to real images) and IS (Salimans et al., 2016) (image quality and diversity) as standard generative metrics, alongside human-aligned metrics that assess distinct dimensions of generation quality: the Aesthetic Score Predictor (Schuhmann et al., 2022) (visual appeal), CMMD (Jayasumana et al., 2024) (distributional similarity in CLIP space), ImageReward (Xu et al., 2023) (overall human preference), and CLIP-Score (Hessel et al., 2021) (text–image alignment). We use COCO2014 (Lin et al., 2014) for evaluation. *Further details are provided in Appendix Section F.*

### 5.1. Improvements on Stable Diffusion Series

We evaluate TAG across the Stable Diffusion family under both unconditional and conditional generation settings. Table 1 presents unconditional results on widely used backbones. *Without requiring additional #NFEs*, TAG consistently improves FID and CMMD across all backbones. This improvement extends to the conditional setting: Table 2 reports T2I results on COCO2014, where TAG consistently improves both sample fidelity and prompt alignment across diverse measures. Qualitatively, Figure 6 corroborates these findings: TAG suppresses artifacts and produces more coherent structures, resolving blob-like formations and implausible anatomy (e.g., a woman with three legs), and yielding sharper, cleaner compositions (e.g., the dog-and-cat scene).

**Compatibility with existing guidance methods.** TAG is orthogonal to existing residual-based guidance methods such as PAG and SEG, with an additional SSG compatibility check provided in Appendix Section B.1. Tables 3 and 4 show that stacking TAG on top of these methods further improves sample quality at matched #NFEs, without architectural changes or additional model evaluations. Notably, the gains are consistent across several metrics, especially on human-preference-oriented scores, suggesting that TAG can refine existing guidance outputs without substantially changing the overall distribution. As shown in Figure 6, TAG fixes both an *action-binding* error (*"A girl **flying a red kite**..."*— baseline depicts the girl *as* airborne) and a *numeracy* error (*"**A horse** standing in a field..."*—baseline yields two). *Additional qualitative comparisons are provided in Figures 12 to 14 in Appendix Section C.3.*

*Table 3.* **Quantitative comparison with existing guidance methods for *unconditional* generation** on the SD-v1.5 using COCO 2014.

| Uncond. | FID ↓ | IS ↑ | AES ↑ | CMMD ↓ |
|---|---|---|---|---|
| No guidance | 58.41 | 15.59 | 5.003 | 1.069 |
| **TAG** | **46.20** | **16.77** | **5.064** | **0.778** |
| PAG (Ahn et al., 2024) | 53.72 | 21.13 | 5.303 | 0.723 |
| **TAG + PAG** | **52.61** | **21.20** | **5.305** | **0.701** |
| SEG (Hong, 2024) | 47.69 | 18.50 | **5.084** | 0.835 |
| **TAG + SEG** | **42.71** | **19.45** | 5.076 | **0.746** |

*Table 4.* **Quantitative comparison with existing guidance methods for *conditional* generation** on the SD-XL using COCO 2014.

| Cond. | FID ↓ | IR ↑ | Pick ↑ | CLIP ↑ |
|---|---|---|---|---|
| No CFG | 83.74 | -0.897 | 20.19 | 21.16 |
| **TAG** | **69.87** | **-0.710** | **20.35** | **22.37** |
| CFG + PAG | 30.24 | 0.352 | 22.13 | 25.23 |
| **TAG$_{cfg}$ + PAG** | **30.22** | **0.354** | 22.13 | **25.25** |
| CFG + SEG | 34.47 | 0.354 | 21.97 | 25.08 |
| **TAG$_{cfg}$ + SEG** | **34.05** | **0.376** | **21.99** | **25.15** |

*Table 5.* **Compositional faithfulness evaluation on T2I-CompBench.** TAG is applied to Stable Diffusion XL on the Spatial and Complex subsets, where structural hallucinations are most common.

| Method | Spatial-Val300 | | | | Method | Complex-Val300 | | | | | | |
|---|---|---|---|---|---|---|---|---|---|---|---|---|
| | 2DSpatial ↑ | AES ↑ | CLIP ↑ | IR ↑ | | BLIP-VQA ↑ | 2DSpatial ↑ | CompCLIP ↑ | 3-in-1 ↑ | AES ↑ | CLIP ↑ | IR ↑ |
| SDXL | 0.1857 | **5.779** | 27.365 | 0.800 | SDXL | 0.4443 | **0.0243** | 0.2910 | 0.3364 | 5.666 | 25.975 | 0.2596 |
| **+TAG$_{cfg}$** | **0.1980** | 5.768 | **27.714** | **0.911** | **+TAG$_{cfg}$** | **0.4650** | 0.0232 | **0.2937** | **0.3472** | **5.667** | **26.477** | **0.3978** |

## 5.2. Validation on Hallucination-oriented Benchmark

To further assess whether TAG's gains reflect improved compositional faithfulness rather than altered perceptual style, we evaluate on *T2I-CompBench* (Huang et al., 2023), focusing on the *'Spatial'* and *'Complex'* subsets where structural hallucinations are most prevalent (e.g., incorrect object placement and multi-object relations). As shown in Table 5, TAG applied to SDXL improves key compositional metrics while maintaining comparable aesthetic and CLIP-Scores, suggesting that TAG mitigates structural artifacts in hallucination-prone prompts rather than merely shifting perceptual style.

## 5.3. Comparison with geometry-aware guidance.

A line of work improves T2I generation by modifying the CFG signal. In particular, TCFG (Kwon et al., 2025) and APG (Sadat et al., 2025) exploit geometric structure in the guidance signal to mitigate undesirable effects induced by guidance. To contextualize **TAG$_{cfg}$** within this family, we compare it with related geometry-aware CFG methods. Table 6 reports this comparison under a unified evaluation protocol: **TAG$_{cfg}$** achieves the lowest FID among geometry-aware CFG variants, although APG yields the highest CLIP-Score. These results suggest that tangential trajectory amplification constitutes an effective and distinct form of CFG-signal correction. *Qualitative comparisons are provided in Figures 15 and 16 in Appendix Section C.4.*

## 5.4. Improvements on Modern Image & Video Models

**Image generation.** We further evaluate TAG on Qwen-Image (Wu et al., 2025), a strong contemporary backbone whose structural artifacts have been largely mitigated by data curation and fine-tuning. As shown in Table 7, TAG improves *NR-IQA* metrics (Ke et al., 2021; Wang et al., 2023) alongside a marginal FID increase, suggesting that on a backbone already near the reference distribution, tangential amplification primarily refines perceptual detail rather

*Table 6.* **Quantitative comparison with geometry-aware guidance methods (APG, TCFG)** and TAG on COCO 2014.

| COCO@10K | FID ↓ | CLIPScore ↑ |
|---|---|---|
| TCFG (Kwon et al., 2025) | 20.72 | 25.63 |
| APG[†] (Sadat et al., 2025) | 19.52 | **26.71** |
| **TAG$_{cfg}$** | **19.29** | 26.23 |

[†]**Note.** APG primarily targets oversaturation artifacts in the high-CFG regime. We include it to provide a reference point within geometry-aware CFG methods.

*Table 7.* **TAG on Qwen-Image.** TAG improves no-reference perceptual quality metrics on a strong contemporary backbone.

| COCO@10K | CLIPIQA ↑ | MUSIQ ↑ | FID ↓ |
|---|---|---|---|
| Qwen-I (Wu et al., 2025) | 0.4947 | 67.90 | **57.53** |
| **TAG** | **0.5051** | **68.85** | 59.84 |

than shifting distributional statistics. This confirms that TAG can serve as a complementary refinement even for well-optimized pipelines, improving perceived visual quality with only a small trade-off in FID on the base model.

**Video generation.** To validate TAG beyond image generation, we evaluate it on Wan2.2 (Team Wan et al., 2025), using 100 randomly sampled *VBench* (Huang et al., 2024) prompts. As shown in Table 8, TAG improves over the baseline on most metrics, including *dynamic degree, imaging quality, aesthetic quality, overall consistency, and temporal style*. Notably, the increased dynamic degree alongside quality gains suggests that TAG supports more dynamic video generation without simply compressing motion. As shown in Figure 7, Wan2.2 occasionally produces structural artifacts in scenes with complex object compositions, whereas TAG reliably preserves the coherence of such scenes. *Additional qualitative comparisons are provided in Figure 11 in Appendix Section C.2.*

## 6. Discussion

**Solver-agnostic improvements across ODE solvers.** Table 9 compares TAG with representative stronger samplers for unconditional diffusion sampling on ImageNet using SD v1.4 (Rombach et al., 2022). TAG alone provides a substantial improvement over the baseline and achieves per-

*Table 8.* **Video generation with Wan2.2.** TAG is evaluated on 100 VBench prompts.

| Method | Visual Quality | | | Consistency | | | Temporal | | |
|---|---|---|---|---|---|---|---|---|---|
| | Dyn. Deg.↑ | Img. Qual.↑ | Aes. Qual.↑ | BG Cons.↑ | Subj. Cons.↑ | Overall Cons.↑ | Human Act.↑ | Mot. Smooth↑ | Temp. Style↑ |
| Wan2.2 | 0.52 | 0.7047 | 0.5647 | 0.9644 | 0.9613 | 0.1796 | 0.05 | **0.9871** | 0.1796 |
| **TAG** | **0.56** | **0.7066** | **0.5697** | **0.9672** | 0.9625 | **0.1836** | **0.07** | 0.9864 | **0.1836** |

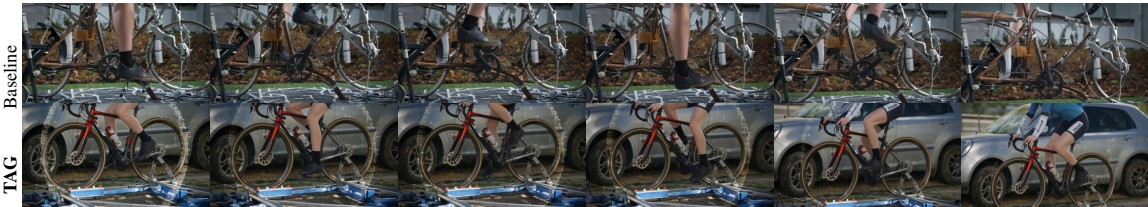

*Figure 7.* **Qualitative comparison of Wan 2.2** baseline (top) and TAG (bottom) on *"A bicycle slowing down to stop."* Six uniformly sampled frames from a 49-frame clip.

*Table 9.* **Comparison with stronger unconditional samplers and their plug-and-play combination with TAG.** All experiments are conducted with Stable Diffusion v1.4 on ImageNet, evaluated by FID@30K and Inception Score (IS).

| Method | FID@30K ↓ | IS ↑ |
|---|---|---|
| Baseline | 65.55 | $14.53 \pm 0.32$ |
| DPM++ (Lu et al., 2023) | 55.40 | $15.92 \pm 0.25$ |
| **TAG** | 52.83 | $16.16 \pm 0.31$ |
| UniPC (Zhao et al., 2023) | 50.82 | $16.37 \pm 0.25$ |
| ▶ **Plug-and-Play** | | |
| UniPC + **TAG** | 48.34 | $16.86 \pm 0.36$ |
| DPM++ + **TAG** | **44.08** | $\mathbf{17.77 \pm 0.36}$ |

*Table 10.* **Computational Cost Analysis.** Measurements were conducted using the DeepSpeed Profiler. FLOPs represents the estimated floating-point operations per image.

| Cost | Param. | FLOPs | VRAM | Latency | Overhead |
|---|---|---|---|---|---|
| Baseline | 0.86 B | 40.99 T | 2.61 GB | 0.919 s | N/A |
| PAG | 0.86 B | $\approx 82$ T | 4.95 GB | 1.956 s | +112.8% |
| **TAG** | 0.86 B | $\approx 41$ T | 2.61 GB | 1.005 s | +9.4% |

formance competitive with widely used sampler variants such as DPM++ (Lu et al., 2023) and UniPC (Zhao et al., 2023). Furthermore, applying TAG on top of these samplers yields additional gains in both FID and IS. Together, these results position TAG as a lightweight refinement of the sampling update that is both competitive as a standalone method and complementary in a plug-and-play manner to stronger solver variants.

**Efficiency and overhead.** We evaluate the computational implications of TAG by reporting FLOPs, peak memory, and wall-clock inference time. All measurements are obtained on an NVIDIA RTX 4090 using the DeepSpeed Profiler (Rajbhandari et al., 2020). As shown in Table 10, diffusion sampling is largely dominated by the UNet forward pass. PAG (Ahn et al., 2024), which introduces a perturbed attention branch and requires an extra UNet evaluation per sampling step, substantially increasing compute, memory, and latency. In contrast, TAG is applied directly to the solver update computed from the same UNet pass and introduces only lightweight vector projections. Empirically, TAG matches baseline peak memory and adds only marginal inference-time overhead while improving sample quality.

*Table 11.* **Effect of timestep windowing in TAG.** TAG is activated only during a timestep window, motivated by the Gaussian annulus theorem. We evaluate FID/IS using 30K ImageNet *val* samples with Stable Diffusion v1.5, DDIM sampler (Song et al., 2021a).

| Uncond. | $[\eta_{\mathrm{sta}}, \eta_{\mathrm{end}}]$ | scale $\eta$ | #NFEs | FID ↓ | IS ↑ |
|---|---|---|---|---|---|
| **TAG**$_{\text{SD v1.5}}$ | [400, 0] | 1.15 | 50 | 69.428 | 16.104 |
| **TAG**$_{\text{SD v1.5}}$ | [1000,0] | 1.15 | 50 | 67.805 | 16.487 |
| **TAG**$_{\text{SD v1.5}}$ | [1000,400] | 1.15 | 50 | **63.870** | **17.516** |

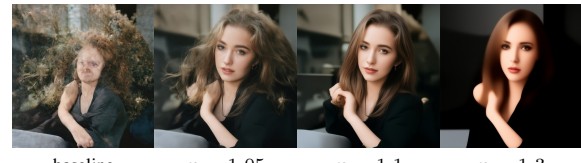

| baseline | $\eta = 1.05$ | $\eta = 1.1$ | $\eta = 1.3$ |
|---|---|---|---|

*Figure 8.* **Ablation on TAG amplification $\eta$.** Moderate $\eta$ improves detail and coherence, while excessive $\eta$ reduces fidelity.

### 6.1. $\eta$ Selection and Adaptive Scaling

The amplification factor $\eta$ controls the strength of the tangential correction in TAG. As illustrated in Figure 8, moderate amplification improves structural detail and visual coherence, whereas larger values reduce fidelity and produce overly smoothed or distorted outputs.

A simple practical way to mitigate this sensitivity is to restrict TAG to a selected timestep interval, specified by $[\eta_{\mathrm{sta}}, \eta_{\mathrm{end}}]$ in Table 11, rather than applying the same amplification over the entire sampling trajectory. In particular, activating TAG mainly in the high-to-intermediate noise regime can avoid over-amplifying late denoising steps, where small tangential perturbations more directly affect visible image details. However, such timestep windowing still requires manual tuning and is model- and sampler-dependent, and therefore remains heuristic.

This non-monotone behavior motivates treating $\eta$ as a sensitivity parameter rather than a simple improvement knob, and highlights the need for adaptive scaling, described below.

**Adaptive Scaling for Robust Amplification.** A practical challenge in TAG is that the effect of $\eta$ depends not only on its nominal value, but also on the instantaneous magnitude of the tangential correction. Since $\|g_k^{\text{align}}\|$ can vary

*Table 12.* **Fixed vs. adaptive scaling of $\eta$ in TAG.** FID@10K on COCO for conditional generation.

| COCO | Baseline | $\eta = 1.0$ | $\eta = 2.0$ | $\eta = 3.0$ |
|---|---|---|---|---|
| Fixed | 28.24 | **25.39** | 26.61 | 29.63 |
| Adaptive | 28.24 | 26.34 | **25.90** | **25.86** |

substantially across timesteps, a fixed global $\eta$ may produce inconsistent correction strengths. Recall the TAG update at step $k$ (Equation (26))

$$\tilde{\varepsilon}_k = \varepsilon_u + \omega\, \boldsymbol{g}_k + \eta\, \boldsymbol{g}_k^{\text{align}}. \tag{27}$$

To reduce this sensitivity, we normalize the tangential term relative to the original guidance scale:

$$\hat{\boldsymbol{g}}_k^{\text{align}} = \frac{\|\boldsymbol{g}_k\|}{\|\boldsymbol{g}_k^{\text{align}}\| + \delta}\, \boldsymbol{g}_k^{\text{align}}. \tag{28}$$

where $0 < \delta \ll 1$ ensures numerical stability. Under this formulation, $\eta$ controls the *relative* amplification of the tangential term with respect to $\|\boldsymbol{g}_k\|$, making the update robust to timestep-dependent variation in $\|\boldsymbol{g}_k^{\text{align}}\|$. As shown in Table 12, adaptive scaling maintains stable performance across a wider range of $\eta$, whereas fixed scaling degrades rapidly at larger values.

# 7. Conclusion, Limitations, and Implications

This paper offers a geometric view of hallucinations in diffusion models by decomposing each sampling update into normal and tangential components. We find that the tangential component is closely tied to semantic organization, motivating a simple intervention on this direction. Building on this insight, we propose **T**angential **A**mplifying **G**uidance (**TAG**), *a theoretically grounded, computationally lightweight, architecture-agnostic* method that amplifies the tangential component during sampling. TAG steers trajectories toward higher-density regions of the data manifold, producing samples with fewer hallucinations and improved fidelity.

**Limitations.** Our theoretical analysis establishes local first-order likelihood gain, but does not fully characterize the behavior of the reverse trajectory. A simple radial–tangential approximation enables efficient guidance, yet may not capture the full geometry of complex data manifolds.

**Implications and Future work.** Despite these simplifications, our results suggest viewing diffusion guidance as geometric shaping of the sampling trajectory. They also indicate that tangential components can serve as semantic refinement signals, motivating future trajectory-level analyses and lightweight estimates of local manifold geometry.

# Acknowledgments

This work was supported in part by the National Research Foundation of Korea (NRF) grant funded by the Korea government (MSIT) (RS-2024-00335741), the Institute of Information & Communications Technology Planning & Evaluation (IITP) grant funded by the Korea government (MSIT) (RS-2025-25442405, Development of a Self-Learning World Model-Based AGI System for Hyperspectral Imaging), the Culture, Sports and Tourism R&D Program through the Korea Creative Content Agency grant funded by the Ministry of Culture, Sports and Tourism (RS-2024-00345025, International Collaborative Research and Global Talent Development for the Development of Copyright Management and Protection Technologies for Generative AI), and the AI Computing Infrastructure Enhancement (GPU Rental Support) User Support Program funded by the Ministry of Science and ICT (MSIT), Republic of Korea (RQT-25-120217).

# Impact Statement

This paper presents Tangential Amplifying Guidance (TAG), an inference-time modification to diffusion sampling that improves fidelity with minimal additional computation. TAG may be useful in applications such as creative tools, prototyping, and education, but higher-fidelity generation can also increase risks of misuse, including deceptive content. Since TAG builds on existing models, it may inherit their biases and limitations. Deployments should use appropriate safety, legal, and licensing safeguards.

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

## A. Proof & Derivation

**Proof for Theorem 4.1**

*Proof.* Assume the deterministic base step $\Delta_{k+1} = \tilde{\alpha}_k\, \epsilon_\theta(\boldsymbol{x}_{k+1}, t_{k+1}) + \beta_k\, \boldsymbol{x}_{k+1}$, with $\tilde{\alpha}_k \leq 0$, and let $\boldsymbol{P}_{\mathcal{M}_{k+1}}, \boldsymbol{P}^\perp_{\mathcal{M}_{k+1}}$ be the orthogonal projectors with $\boldsymbol{P}_{\mathcal{M}_{k+1}} \boldsymbol{x}_{k+1} = \boldsymbol{x}_{k+1}$ and $\boldsymbol{P}^\perp_{\mathcal{M}_{k+1}} \boldsymbol{x}_{k+1} = \boldsymbol{0}$. Applying the projectors to the base decomposition gives

$$\boldsymbol{P}^\perp_{\mathcal{M}_{k+1}} \Delta_{k+1} = \tilde{\alpha}_k\, \boldsymbol{P}^\perp_{\mathcal{M}_{k+1}} \epsilon_\theta(\boldsymbol{x}_{k+1}, t_{k+1}), \tag{29}$$

$$\boldsymbol{P}_{\mathcal{M}_{k+1}} \Delta_{k+1} = \tilde{\alpha}_k\, \boldsymbol{P}_{\mathcal{M}_{k+1}} \epsilon_\theta(\boldsymbol{x}_{k+1}, t_{k+1}) + \beta_k\, \boldsymbol{x}_{k+1}. \tag{30}$$

Therefore, the TAG update rule step is

$$\Delta^{\text{TAG}}_{k+1} = \left( \boldsymbol{P}_{\mathcal{M}_{k+1}} + \eta\, \boldsymbol{P}^\perp_{\mathcal{M}_{k+1}} \right) \Delta_{k+1} = \tilde{\alpha}_k \left[ \boldsymbol{P}_{\mathcal{M}_{k+1}} + \eta \boldsymbol{P}^\perp_{\mathcal{M}_{k+1}} \right] \epsilon_\theta(\boldsymbol{x}_{k+1}, t_{k+1}) + \beta_k \boldsymbol{x}_{k+1}. \tag{31}$$

The first-order Taylor gain with respect to TAG update at $t_{k+1}$ is defined as:

$$\begin{aligned} G(\eta) &:= \left( \Delta^{\text{TAG}}_{k+1} \right)^\top \nabla_{\boldsymbol{x}} \log p(\boldsymbol{x} \mid t_{k+1}) \big|_{\boldsymbol{x}=\boldsymbol{x}_{k+1}} \\ &= \left( \left( \boldsymbol{P}_{\mathcal{M}_{k+1}} + \eta\, \boldsymbol{P}^\perp_{\mathcal{M}_{k+1}} \right) \Delta_{k+1} \right)^\top \nabla_{\boldsymbol{x}} \log p(\boldsymbol{x} \mid t_{k+1}) \big|_{\boldsymbol{x}=\boldsymbol{x}_{k+1}} \end{aligned} \tag{32}$$

We analyze this gain by approximating the true score with the model's score function

$$\boldsymbol{s}_\theta(\boldsymbol{x}_{k+1}, t_{k+1}) = -\sigma^{-1}_{k+1} \epsilon_\theta(\boldsymbol{x}_{k+1}, t_{k+1}), \tag{33}$$

thus:

$$\begin{aligned} G(\eta) &= \left( \left( \boldsymbol{P}_{\mathcal{M}_{k+1}} + \eta\, \boldsymbol{P}^\perp_{\mathcal{M}_{k+1}} \right) \Delta_{k+1} \right)^\top \nabla_{\boldsymbol{x}} \log p(\boldsymbol{x} \mid t_{k+1}) \big|_{\boldsymbol{x}=\boldsymbol{x}_{k+1}} \\ &\approx \left( \left( \boldsymbol{P}_{\mathcal{M}_{k+1}} + \eta\, \boldsymbol{P}^\perp_{\mathcal{M}_{k+1}} \right) \Delta_{k+1} \right)^\top \boldsymbol{s}_\theta(\boldsymbol{x}_{k+1}, t_{k+1}) \\ &= -\sigma^{-1}_{k+1} \cdot \left( \left( \boldsymbol{P}_{\mathcal{M}_{k+1}} + \eta\, \boldsymbol{P}^\perp_{\mathcal{M}_{k+1}} \right) \Delta_{k+1} \right)^\top \epsilon_\theta(\boldsymbol{x}_{k+1}, t_{k+1}) \end{aligned} \tag{34}$$

Substitute Equation (31) into Equation (34), then:

$$\begin{aligned} G(\eta) &\approx -\sigma^{-1}_{k+1} \left( \tilde{\alpha}_k \boldsymbol{P}_{\mathcal{M}_{k+1}} \epsilon_\theta + \beta_k \boldsymbol{x}_{k+1} + \eta \tilde{\alpha}_k \boldsymbol{P}^\perp_{\mathcal{M}_{k+1}} \epsilon_\theta \right)^\top \epsilon_\theta \\ &= -\sigma^{-1}_{k+1} \left( \tilde{\alpha}_k (\boldsymbol{P}_{\mathcal{M}_{k+1}} \epsilon_\theta)^\top \epsilon_\theta + \beta_k \boldsymbol{x}^\top_{k+1} \epsilon_\theta + \eta \tilde{\alpha}_k (\boldsymbol{P}^\perp_{\mathcal{M}_{k+1}} \epsilon_\theta)^\top \epsilon_\theta \right). \end{aligned} \tag{35}$$

Since $\boldsymbol{P}$ and $\boldsymbol{P}^\perp$ are symmetric and idempotent, thus

$$\boldsymbol{v}^\top \boldsymbol{P} \boldsymbol{v} = \| \boldsymbol{P} \boldsymbol{v} \|^2_2 \tag{36}$$

is established. Therefore,

$$G(\eta) \approx -\sigma^{-1}_{k+1} \left( \tilde{\alpha}_k \| \boldsymbol{P}_{\mathcal{M}_{k+1}} \epsilon_\theta \|^2_2 + \beta_k \boldsymbol{x}^\top_{k+1} \epsilon_\theta + \eta \tilde{\alpha}_k \| \boldsymbol{P}^\perp_{\mathcal{M}_{k+1}} \epsilon_\theta \|^2_2 \right). \tag{37}$$

Differentiating the gain $G(\eta)$ in Equation (37) with respect to $\eta$ yields:

$$\frac{\partial G(\eta)}{\partial \eta} \approx \frac{-\tilde{\alpha}_k}{\sigma_{k+1}} \left| \boldsymbol{P}^\perp_{\mathcal{M}_{k+1}} \epsilon_\theta(\boldsymbol{x}_{k+1}, t_{k+1}) \right|^2_2 \geq 0. \tag{38}$$

This derivative is guaranteed to be non-negative, since the DDIM sampler coefficient $\tilde{\alpha}_k \leq 0$ by definition, while $\sigma_{k+1}$ and the squared L2-norm are strictly non-negative. This proves that the first-order gain $G(\eta)$ is a monotonically non-decreasing function of $\eta$. Consequently, amplifying the tangential component of the update step via TAG is guaranteed to improve the first-order log-likelihood gain compared to the base update step.

**Analysis on pure TAG gain.** Subtracting each gain $G^{\text{base}} \triangleq G(\eta = 1)$ and $G^{\text{TAG}} \triangleq G(\eta > 1)$,

$$\overbrace{\left( - \sigma_{k+1}^{-1} \cdot \left(\Delta_{k+1}^{\text{TAG}}\right)^{\top} \epsilon_\theta\big(\boldsymbol{x}_{k+1}, t_{k+1}\big) \right)}^{\text{TAG update gain, } G^{\text{TAG}}} - \overbrace{\left( - \sigma_{k+1}^{-1} \cdot \left(\Delta_{k+1}\right)^{\top} \epsilon_\theta\big(\boldsymbol{x}_{k+1}, t_{k+1}\big) \right)}^{\text{base update gain, } G^{\text{base}}}$$

$$= -\sigma_{k+1}^{-1} \cdot \left(\Delta_{k+1}^{\text{TAG}} - \Delta_{k+1}\right)^{\top} \epsilon_\theta\big(\boldsymbol{x}_{k+1}, t_{k+1}\big)$$

$$= -\sigma_{k+1}^{-1} \cdot \left((\eta - 1)\, \boldsymbol{P}_{\mathcal{M}_{k+1}}^{\perp} \Delta_{k+1}\right)^{\top} \epsilon_\theta\big(\boldsymbol{x}_{k+1}, t_{k+1}\big). \tag{39}$$

Using $\Delta_{k+1} = \tilde{\alpha}_k\, \epsilon_\theta(\boldsymbol{x}_{k+1}, t_{k+1}) + \beta_k\, \boldsymbol{x}_{k+1}$, $\boldsymbol{P}_{\mathcal{M}_{k+1}}^{\perp}$ be:

$$\boldsymbol{P}_{\mathcal{M}_{k+1}}^{\perp} \Delta_{k+1} = \boldsymbol{P}_{\mathcal{M}_{k+1}}^{\perp} \tilde{\alpha}_k\, \epsilon_\theta(\boldsymbol{x}_{k+1}, t_{k+1}). \tag{40}$$

Thus, substitute Equation (40) into Equation (39) then:

$$G^{\text{TAG}} - G^{\text{base}} = \underbrace{-\sigma_{k+1}^{-1} \cdot \left(\tilde{\alpha}_k(\eta - 1)\right)}_{\text{scalar}} \cdot \left(\boldsymbol{P}_{\mathcal{M}_{k+1}}^{\perp} \epsilon_\theta(\boldsymbol{x}_{k+1}, t_{k+1})\right)^{\top} \epsilon_\theta\big(\boldsymbol{x}_{k+1}, t_{k+1}\big). \tag{41}$$

This simplifies to the final quadratic form:

$$G^{\text{TAG}} - G^{\text{base}} = \underbrace{-\sigma_{k+1}^{-1} \cdot \left(\tilde{\alpha}_k(\eta - 1)\right)}_{\geq \, \boldsymbol{0} \;\; \text{as} \;\; \tilde{\alpha}_k \, \leq \, 0} \cdot \left\|\boldsymbol{P}_{\mathcal{M}_{k+1}}^{\perp} \epsilon_\theta(\boldsymbol{x}_{k+1}, t_{k+1})\right\|_2^2, \tag{42}$$

This proves that the difference in gain is non-negative for any $\eta \geq 1$. Therefore, the first-order log-likelihood gain of the TAG update is always greater than or equal to that of the base update, with equality holding if and only if $\eta = 1$ or the tangential component of the score is zero. $\square$

*Table 13.* **Compatibility with Self-Swap Guidance (SSG).** TAG is applied on top of SSG under matched evaluation settings on COCO 2014. TAG consistently improves SSG in both unconditional and conditional generation.

| (a) *Unconditional* generation on SD-v1.5 | | | | | (b) *Conditional* generation on SD-XL | | | | |
|---|---|---|---|---|---|---|---|---|---|
| **Uncond.** | **FID** ↓ | **IS** ↑ | **AES** ↑ | **CMMD** ↓ | **Cond.** | **FID** ↓ | **IR** ↑ | **Pick** ↑ | **CLIP** ↑ |
| SSG (Zhang et al., 2026) | 49.02 | 18.63 | 5.186 | 0.654 | CFG + SSG | 32.82 | 0.449 | **22.40** | 25.11 |
| **TAG + SSG** | **48.53** | **19.05** | **5.187** | **0.648** | $\text{TAG}_{\text{cfg}}$ + SSG | **32.81** | **0.460** | **22.40** | **25.15** |

## B. Broader Experiments

### B.1. Application to Self-Swap Guidance (SSG, Section 5.1)

We further evaluate the compatibility of TAG with Self-Swap Guidance (SSG) (Zhang et al., 2026). SSG constructs a perturbed prediction by swapping token and channel positions in self-attention activations, whereas TAG is applied after the scheduler step by decomposing the resulting update into radial and tangential components. Therefore, TAG can be directly stacked on top of SSG without modifying the model architecture or requiring additional denoising evaluations.

In the unconditional setting, TAG improves SSG across all four metrics, reducing FID from 49.02 to 48.53 and CMMD from 0.654 to 0.648. In the conditional setting, $\text{TAG}_{\text{cfg}}$ also improves FID, ImageReward, and CLIPScore on top of SSG, while maintaining the same PickScore. These results support that TAG is complementary to SSG, as it refines the solver trajectory after SSG guidance rather than replacing the guidance signal itself. Qualitative comparisons are provided in Figure 14.

### B.2. Unconditional Generation on ImageNet.

We further evaluate TAG on unconditional generation to verify that the proposed method generalizes beyond class-conditional and text-conditional settings. Table 14 reports FID-50K on ImageNet $64\times64$ using EDM2 (Karras et al., 2024b) as the baseline. Applying TAG reduces FID from 11.04 to **10.37**, confirming that TAG provides consistent gains even without explicit conditioning signals.

*Table 14.* Unconditional generation on ImageNet $64\times64$ (FID-50K).

| **Method** | $[\eta_{\text{sta}}, \eta_{\text{end}}]$ | **scale** $\eta$ | **FID** ↓ |
|---|---|---|---|
| EDM2 | – | – | 11.0432 |
| **TAG** | [750, 500] | 1.3 | **10.3741** |

### B.3. Beyond 2D Image: Image-to-3D shape generation

To check whether TAG can be applied beyond 2D image synthesis, we run a small qualitative test on image-to-3D generation with SV3D (Voleti et al., 2024). We apply TAG as a lightweight modification during SV3D sampling, without changing the model or adding extra evaluations. As shown in Figure 9, TAG yields modest qualitative improvements, with fewer view-dependent artifacts and slightly more consistent structure across viewpoints. A systematic quantitative study for 3D generation is left for future work.

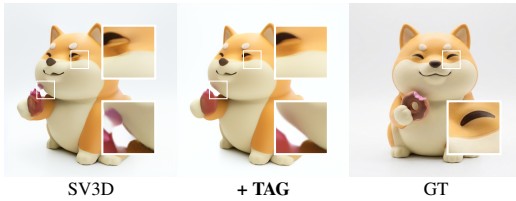

SV3D      + TAG      GT

*Figure 9.* **Qualitative comparison** on SV3D (Voleti et al., 2024). Adding TAG yields modest improvements in cross-view consistency.

# C. Additional Qualitative Results

## C.1. Image Generation: unconditional

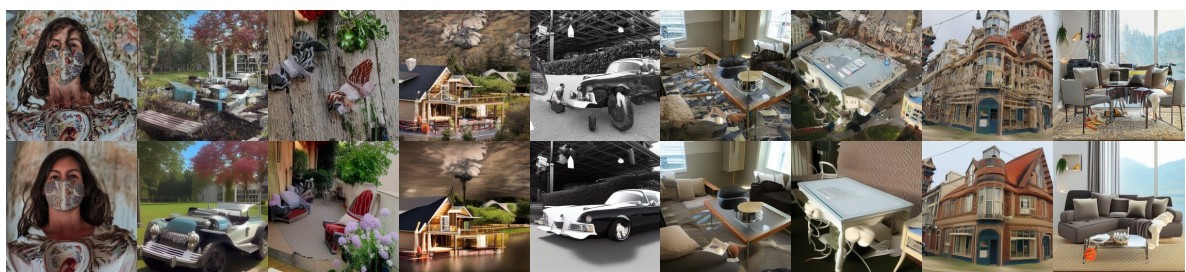

*Stable Diffusion 1.5*

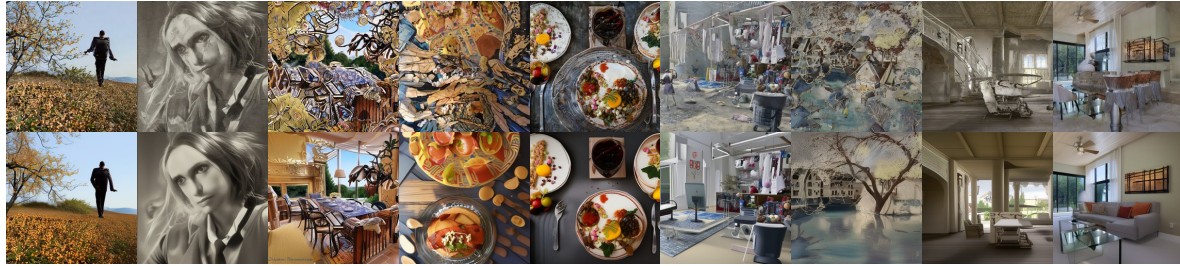

*Stable Diffusion 2.1*

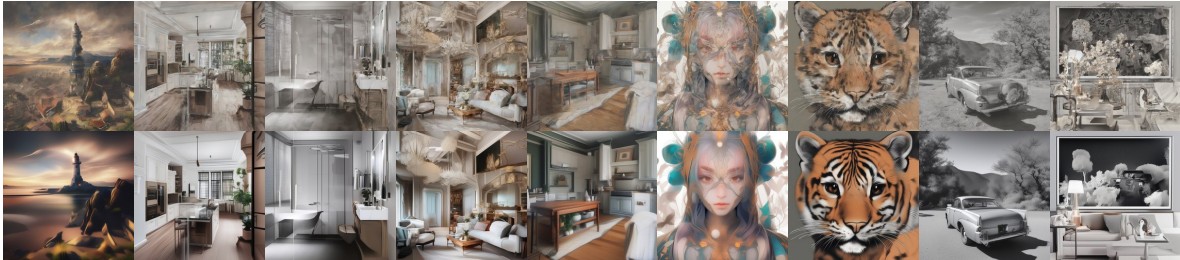

*Stable Diffusion XL*

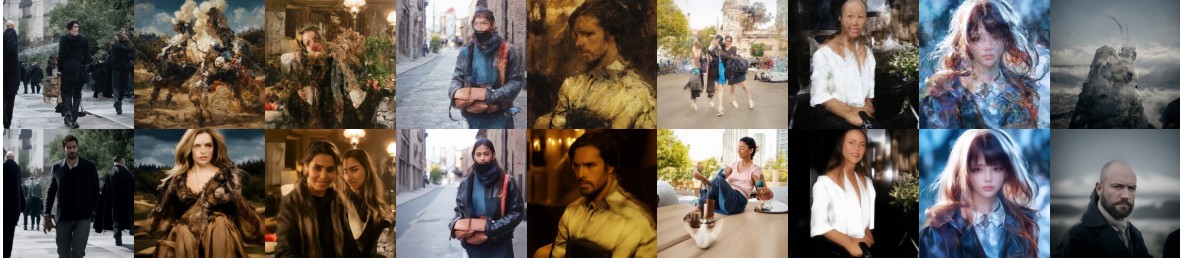

*Stable Diffusion 3*

*Figure 10.* **Qualitative results for unconditional generation across backbones.** For each model (SD v1.5/2.1/XL (Rombach et al., 2022; Podell et al., 2024), and 3 (Esser et al., 2024)), the top row shows baseline sampling and the bottom row shows +TAG at matched NFEs. TAG yields sharper, more coherent structure with fewer artifacts while preserving diversity.

## C.2. Video Generation (Section 5.4)

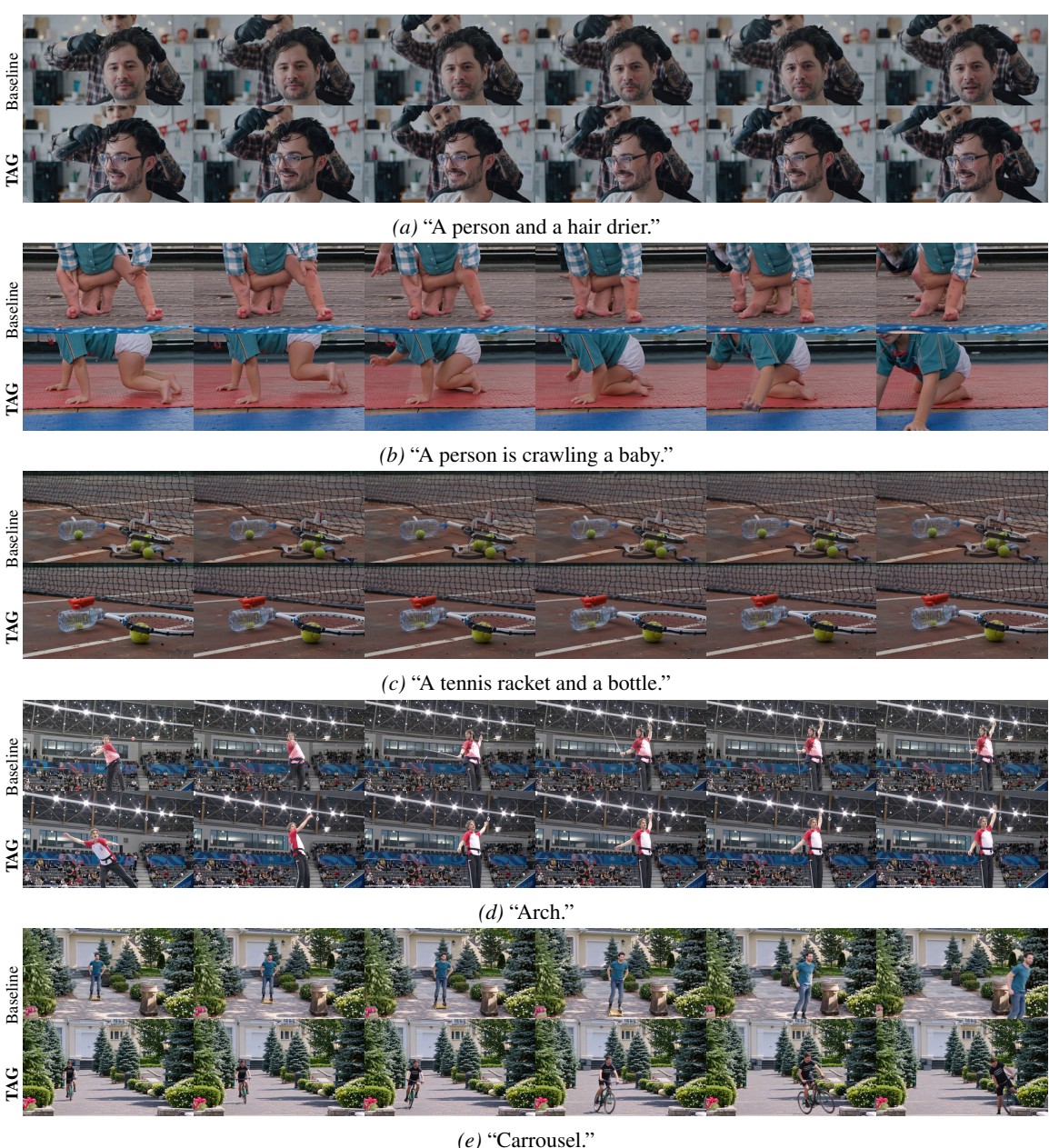

*(a)* "A person and a hair drier."

*(b)* "A person is crawling a baby."

*(c)* "A tennis racket and a bottle."

*(d)* "Arch."

*(e)* "Carrousel."

*Figure 11.* Qualitative comparison between the Wan 2.2 (Team Wan et al., 2025) baseline and our TAG model on five prompts. For each prompt, the top row shows the baseline and the bottom row shows TAG (ours). Six frames are uniformly sampled across each 49-frame clip.

### C.3. Image Generation: Plug-and-Play (for Section 5.1)

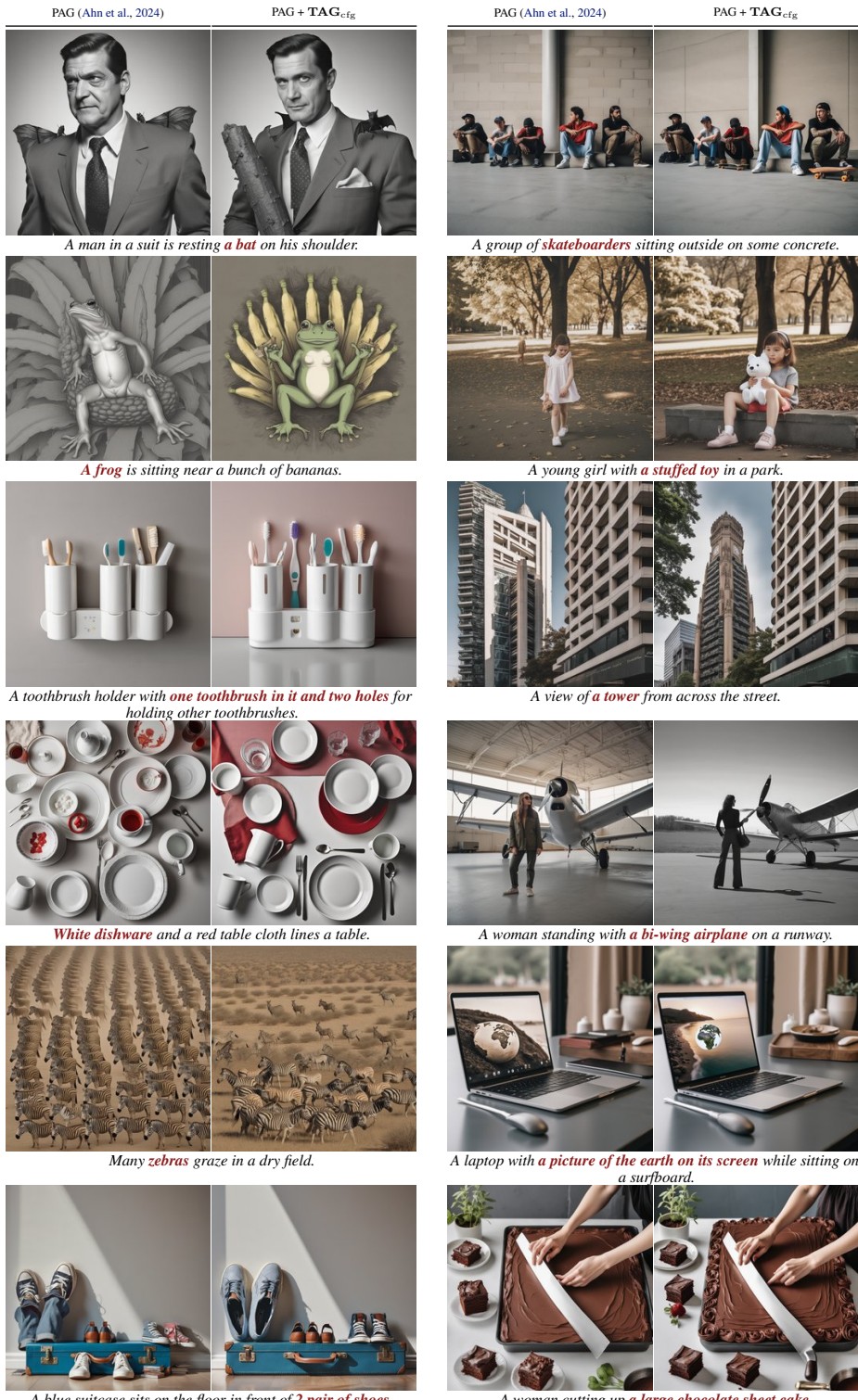

*Figure 12.* **Qualitative comparison between PAG and PAG +** $\mathrm{TAG}_{\mathrm{cfg}}$. The **highlighted words** in each prompt indicate target concepts, attributes, or relations that are often weakly represented or missing in the baseline results. PAG + $\mathrm{TAG}_{\mathrm{cfg}}$ more faithfully reflects these prompt-specific details, such as adding missing objects and correcting actions, while preserving the overall scene structure.

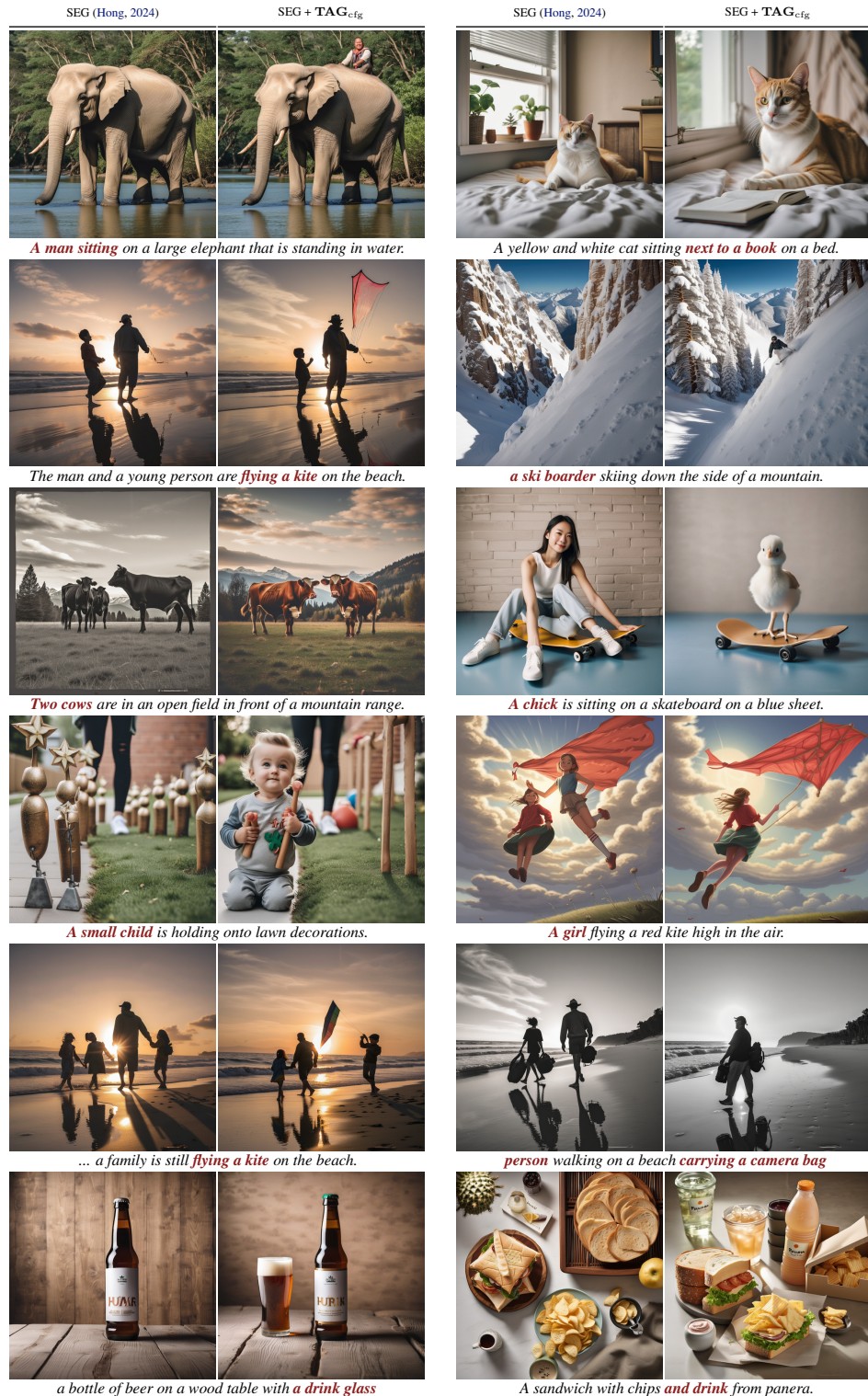

*Figure 13.* **Qualitative comparison between SEG and SEG +** $\mathrm{TAG_{cfg}}$. The *highlighted words* in each prompt indicate target concepts, attributes, or relations that are often weakly represented or missing in the baseline results. SEG + $\mathrm{TAG_{cfg}}$ more faithfully reflects these prompt-specific details, such as adding missing objects and correcting actions, while preserving the overall scene structure.

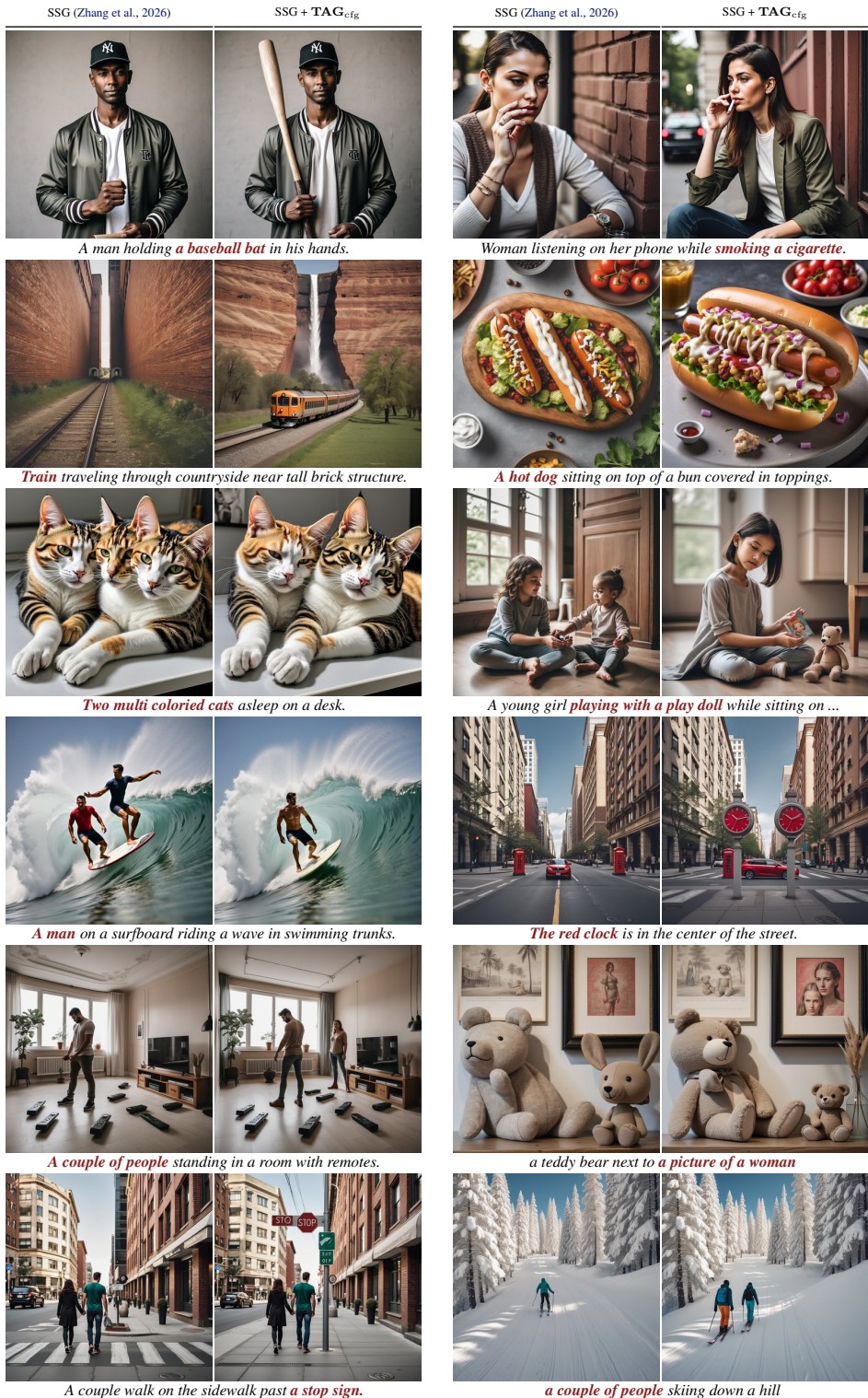

*Figure 14.* **Qualitative comparison between SSG and SSG +** $\text{TAG}_{\text{cfg}}$. The *highlighted words* in each prompt indicate target concepts, attributes, or relations that are often weakly represented or missing in the baseline results. SSG + $\text{TAG}_{\text{cfg}}$ more faithfully reflects these prompt-specific details, such as adding missing objects and correcting actions, while preserving the overall scene structure.

## C.4. Image Generation: Geometric Guidance (for Section 5.3)

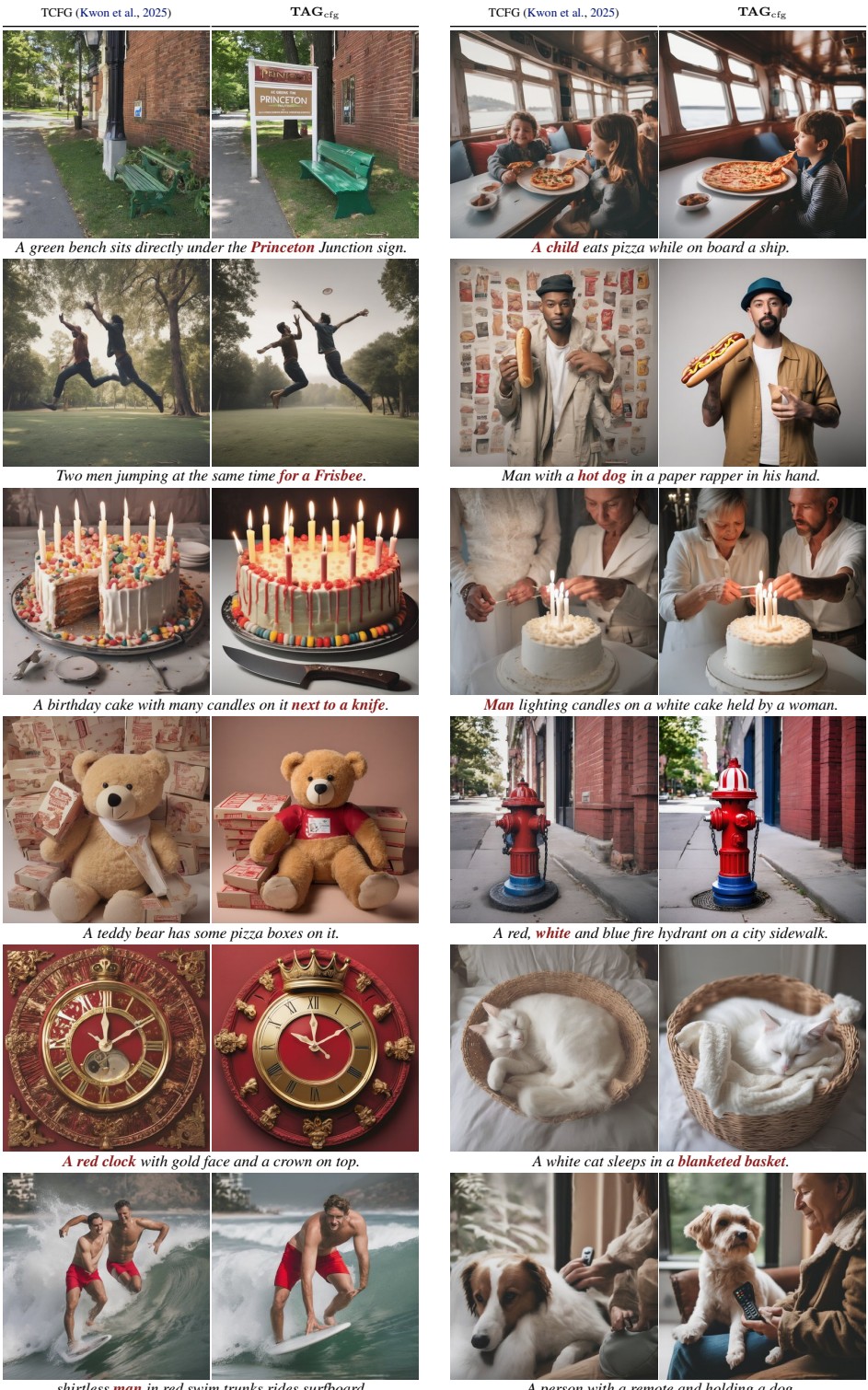

*Figure 15.* **Qualitative comparison between TCFG and** $\text{TAG}_{\text{cfg}}$. The ***highlighted words*** in each prompt indicate target concepts, attributes, or relations that are often weakly represented or missing in the baseline results. $\text{TAG}_{\text{cfg}}$ more faithfully reflects these prompt-specific details, such as adding missing objects and correcting actions, while preserving the overall scene structure.

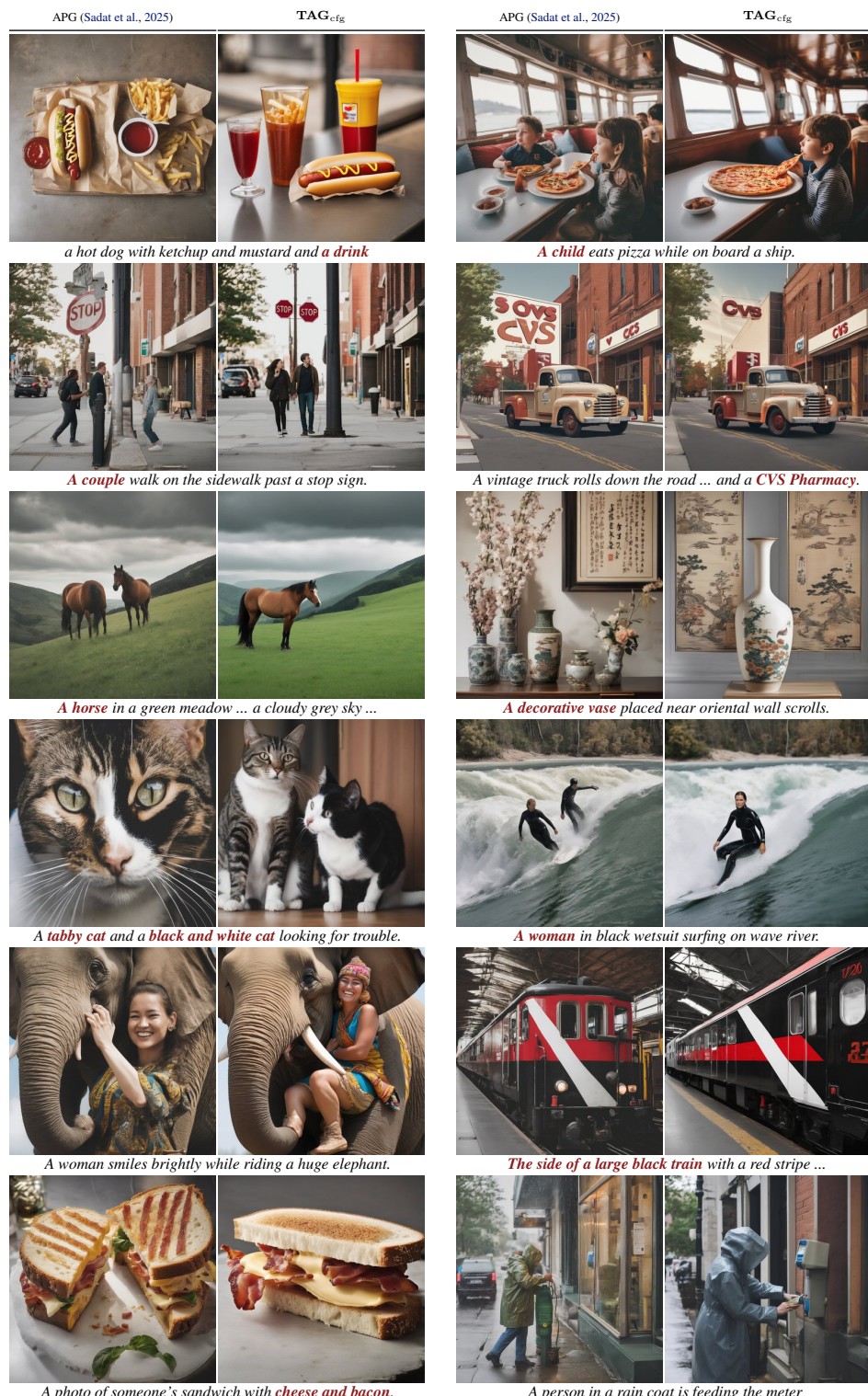

*Figure 16.* **Qualitative comparison between APG and** TAG$_{cfg}$**.** The ***highlighted words*** in each prompt indicate target concepts, attributes, or relations that are often weakly represented or missing in the baseline results. TAG$_{cfg}$ more faithfully reflects these prompt-specific details, such as adding missing objects and correcting actions, while preserving the overall scene structure.

# D. Implementation of the Tangential Amplifying Guidance (Section 4.2)

**Algorithm 2** Code: Tangential Amplifying Guidance (TAG)

```
output = scheduler.step(noise_pred, t, latents, return_dict=False)

if apply_tag:
    post = latents
    eta_v, eta_n = t_guidance_scale, 1

    v_t = post / (post.norm(p=2, dim=(1,2,3), keepdim=True) + 1e-8)

    latents = output
    delta = latents - post
    a     = (delta * v_t).sum(dim=(1,2,3), keepdim=True)

    u_n = a * v_t
    u_t = delta - u_n
    latents = post + eta_v * u_t + eta_n * u_n
else:
    latents = output
```

**Algorithm 3** Code: Conditional Tangential Amplifying Guidance (**TAG**$_{cfg}$)

```
def proj_par(z, n):
    return (z * n).sum(dim=(1,2,3), keepdim=True) * n

def proj(z, v):
    v = v / (v.norm(p=2, dim=(1,2,3), keepdim=True) + 1e-8)
    return (z * v).sum(dim=(1,2,3), keepdim=True) * v

eps_u, eps_c = HeadToEps(noise_pred, latents, t, scheduler, do_cfg)

s_u = -eps_u / (sigma + 1e-12)
s_c = -eps_c / (sigma + 1e-12)

n = latents / (latents.norm(p=2, dim=(1,2,3), keepdim=True) + 1e-8)

g       = s_c - s_u
t_c     = s_c - proj_par(s_c, n)
t_u     = s_u - proj_par(s_u, n)
g_aligned = proj(s_c, t_c - t_u)

s_star = s_u + (guidance_scale * g) + (t_guidance_scale * g_aligned)
eps    = -sigma * s_star

model_out = EpsToHead(eps, latents, t, scheduler)
latents = scheduler.step(model_out, t, latents, return_dict=False)
```

# E. Discussion on Orthogonal and Projection-based Guidance (Section 2)

Several recent works incorporate projections or tangential/orthogonal notions in diffusion guidance. While they share surface-level similarity, they differ in (i) *which quantity is decomposed*, (ii) how the relevant *subspace is defined*, and (iii) the *problem setting* and objective that motivate the projection. We summarize these distinctions below to situate TAG.

## E.1. Orthogonal and Projection-based Diffusion Guidance

### E.1.1. CFG-SPECIFIC PROJECTIONS FOR ARTIFACT SUPPRESSION

**Projected CFG for oversaturation.** Sadat et al. (2025) analyze artifacts such as oversaturation at large CFG (Ho & Salimans, 2021) scales. They decompose the CFG update into components parallel and orthogonal to the *conditional* score, and empirically identify the *parallel component as the primary cause of saturation*.

Their Adaptive Projected Guidance (APG) therefore *attenuates the parallel component* while preserving the orthogonal component, and complements this with rescaling and momentum-style heuristics motivated by a gradient-ascent interpretation of CFG. This line of work is *(i) tied to the CFG algebra* and targets *(ii) large-scale saturation* artifacts.

**SVD-based tangential damping within CFG.** Kwon et al. (2025) address conditional-unconditional misalignment in CFG, which can lead to off-manifold samples. They form a score matrix from conditional and unconditional scores and perform SVD, interpreting high singular-value directions as shared, approximately manifold-normal components and *low singular-value directions as tangential components*.

They observe that the discrepancy between conditional and unconditional scores is concentrated in low-SV (i.e., tangential) directions, and propose filtering the *unconditional* score by *projecting it onto the shared high-SV subspace* before applying CFG. This approach modifies only the unconditional term inside CFG rather than the base solver update, and does not study the semantic role of tangential components along the solver trajectory.

### E.1.2. CORRECTION OF CFG DYNAMICS

**Nonlinear correction via characteristics.** Zheng & Lan (2024) derive guidance from the nonlinear Fokker-Planck dynamics of guided diffusion. They show that standard linear *CFG neglects nonlinear correction terms* that become important at high guidance scales, causing artifacts.

Their method solves a nonlinear *fixed-point / characteristic equation* for the corrected guided score. A *projection operator* can be inserted for numerical regularization, but it is not the conceptual driver of the method; the theoretically *ideal operator is the identity*. Thus, improvements come from nonlinear correction *rather than from projecting tangential/orthogonal components*.

### E.1.3. PROMPT INTERACTION AND SEMANTIC DISENTANGLEMENT

**Orthogonalizing negative prompts.** Armandpour et al. (2023) study failures of negative prompting when negative and positive concepts overlap. They decompose the negative score into components *parallel and perpendicular to the positive score* direction, discard the parallel (overlapping) negative component, and apply only the perpendicular negative guidance.

The goal is to prevent negative prompts from canceling shared desirable attributes, especially in text-to-image and text-to-3D pipelines. This mechanism concerns *prompt interaction effects* rather than diffusion-step geometry.

### E.1.4. TASK-SPECIFIC MANIFOLD CONSTRAINTS

**Tangent restriction for loss-based guidance.** He et al. (2024) focus on training-free, loss-based guidance (e.g., *DPS (Chung et al., 2023)-style inverse problems*), noting that ambient-space loss gradients may *violate data-manifold* constraints.

They apply *guidance on $x_{0|t}$* and project external guidance gradients onto tangent spaces of the clean-data manifold $\mathcal{M}_0$, estimating tangents using an auxiliary pretrained autoencoder. These methods rely on an explicit manifold model and are tailored to *observation-conditioned tasks* rather than generic unconditional or standard conditional generation.

**Refinement–transport decomposition in conditional I2I.** In unpaired conditional image-to-image translation, Sun et al. (2023) construct *reference-dependent manifolds* $\mathcal{M}_t(y_0)$ induced by a reference image $y_0$. Such a conditional structure is crucial: the authors emphasize (§4, Lemma 1 of (Sun et al., 2023)) that in standard diffusion, where intermediate manifolds

*Table 15.* **Comparison of projection / tangential guidance methods.** Prior works decompose CFG algebra, prompt gradients, or task-specific manifold scores; TAG uniquely decomposes and amplifies the *single-step solver update* with respect to iso-noise manifolds, without auxiliary models.

| Work | Quantity | Subspace / manifold | Projection / filtering | Goal | Cost |
|---|---|---|---|---|---|
| APG (Sadat et al., 2025) | CFG update | $\parallel$ / $\perp$ to conditional score | Downweight the $\parallel$ component in CFG | Mitigate oversaturation under high CFG | CFG |
| TCFG (Kwon et al., 2025) | Conditional / unconditional scores | SVD shared high-SV vs. tangential low-SV subspaces | Project *unconditional* score to the high-SV subspace before CFG | Reduce conditional–unconditional mismatch in T2I | CFG |
| Characteristic Guidance (Zheng & Lan, 2024) | Guided-score fixed point | None (projection optional) | Optional stabilizing projection $P$; ideal case $P = I$ | Nonlinear correction of CFG | CFG |
| Perp-Neg (Armandpour et al., 2023) | Negative-prompt score | $\parallel$ / $\perp$ to positive prompt | Remove the $\parallel$ negative component | Avoid negative / positive prompt overlap | Multi-prompt |
| MPGD (He et al., 2024) | Loss gradient on $x_{0|t}$ / latent | Tangent space of AE-defined $\mathcal{M}_0$ | Project loss gradient to the tangent space | On-manifold inverse problems | AE extra |
| SDDM (Sun et al., 2023) | I2I conditional score / energy | Reference-induced $\mathcal{M}_t(y_0)$ | Normal transport with tangential refinement | Unpaired image-to-image translation | Task mods |
| TAG (ours) | Solver step $\Delta x_k$ | $\parallel$ / $\perp$ to iso-noise manifold $\mathcal{M}_k$ | Amplify the tangential component of $\Delta x_k$ | Generic sampling refinement | Simple vector calc. |

at adjacent timesteps are coupled, a *tangential/normal score split is generally meaningless*.

In contrast, Sun et al. (2023) argue that the *I2I setting* yields compact, well-separated manifolds across time, making the *decomposition meaningful*; under this specific condition, the tangential component acts as an on-manifold refinement term, while the normal component governs transport between manifolds of adjacent timesteps.

### E.2. TAG: Intrinsic Tangential Amplification for General Diffusion Sampling

TAG leverages the *intrinsic geometry* of diffusion trajectories by decomposing the *single-step solver update* at each state $x_k$ into radial (manifold-normal) and tangential components with respect to the iso-noise manifold $\mathcal{M}_k$. We show that this *split is stable and meaningful* in general diffusion:

- the tangential update captures semantic refinement along $\mathcal{M}_k$

- while the radial part is largely unstructured,

and amplifying the tangential component provably promotes local log-likelihood ascent. In contrast to CFG-algebraic projections (Sadat et al., 2025; Kwon et al., 2025), nonlinear CFG corrections (Zheng & Lan, 2024), prompt-specific orthogonalization (Armandpour et al., 2023), or task-/condition-dependent manifold models (Sun et al., 2023; He et al., 2024), TAG operates directly on the base solver update, requires *no auxiliary models* or *extra network passes*, and *applies uniformly* to unconditional and standard conditional generation across modern backbones.

# F. Experimental Details (for Section 5)

## F.1. Implementation Details

All experiments are implemented in PyTorch 2.5.1 with CUDA 12.1, using the `diffusers` 0.35.1 library (von Platen et al., 2022). Each pipeline subclasses the corresponding `diffusers` pipeline and overrides the `__call__` method to inject the TAG decomposition logic after the scheduler step. All experiments are conducted on NVIDIA GeForce RTX 4090 GPUs (24 GB VRAM each).

## F.2. Unconditional Generation

We evaluate TAG on four Stable Diffusion variants (SD1.5/2.1 (Rombach et al., 2022), SDXL (Podell et al., 2024), and SD3 (Esser et al., 2024)).

For all unconditional experiments, we use an empty prompt (`prompt=""`) with classifier-free guidance disabled ($\omega = 0$). We generate 30,000 images per setting using integer seeds from 0 to 29,999. The number of denoising steps is fixed at $T = 50$ for all models. TAG is applied with the radial scaling factor $\eta_r = 1.0$ (unchanged) across all experiments; only the tangential scaling factor $\eta_v$ varies. The TAG application range covers all timesteps ($t_{\text{start}} = 1000$, $t_{\text{end}} = 0$).

*Table 16.* TAG hyperparameters for unconditional generation (Table 1 in the main paper). All settings use $\eta_r = 1.0$, $t_{\text{start}} = 1000$, $t_{\text{end}} = 0$, and $T = 50$ steps.

| Model | $\eta_v$ | $\eta_r$ | $\omega$ | $t_{\text{start}}$ | $t_{\text{end}}$ | **Steps** |
|---|---|---|---|---|---|---|
| SD v1.5 | (TAG off) | 1.0 | 0.0 | – | – | 50 |
| TAG$_{\text{SD v1.5}}$ | 1.15 | 1.0 | 0.0 | 1000 | 0 | 50 |
| SD v2.1 | (TAG off) | 1.0 | 0.0 | – | – | 50 |
| TAG$_{\text{SD v2.1}}$ | 1.15 | 1.0 | 0.0 | 1000 | 0 | 50 |
| SDXL | (TAG off) | 1.0 | 0.0 | – | – | 50 |
| TAG$_{\text{SDXL}}$ | 1.20 | 1.0 | 0.0 | 1000 | 0 | 50 |
| SD3 | (TAG off) | 1.0 | 0.0 | – | – | 50 |
| TAG$_{\text{SD3}}$ | 1.05 | 1.0 | 0.0 | 1000 | 0 | 50 |

Table 16 lists the tangential scaling factor $\eta_v$ (`t_lr`) used for each model. These values were selected based on preliminary sweeps.

## F.3. TAG for Classifier-Free Guidance

For conditional (text-to-image) experiments, we use 10,000 captions randomly sampled from the MS-COCO 2014 (Lin et al., 2014) validation set as text prompts. All images are generated with a fixed seed of 0 and CFG scale $\omega$.

## F.4. Compatibility with Guidance Methods (PAG, SEG, and SSG)

**PAG + TAG.** Perturbed Attention Guidance (PAG) (Ahn et al., 2024) replaces the self-attention map at designated UNet layers with an identity matrix to create a perturbed prediction, then guides generation away from the perturbed output. We combine PAG with TAG by first applying PAG guidance at the noise-prediction level, then applying the TAG radial–tangential decomposition after the scheduler step. Following the official PAG recommendation, we use `pag_scale= 3.0` applied to the mid-block (`m0`) of the UNet.

*Table 17.* PAG / SEG / SSG compatibility experiment settings (unconditional SD v1.5). SSG uses `swap_ratio=0.001` applied to all 16 attention layers (`d0..d5,m0,u0..u8`).

| Setting | Guidance | Scale | Layer | $\eta_v$ | $\eta_r$ | $t_{\text{start}}$ | $t_{\text{end}}$ | $N$ |
|---|---|---|---|---|---|---|---|---|
| **Baseline** | – | – | – | – | – | – | – | 30K |
| ▶ TAG | – | – | – | 1.15 | 1.0 | 1000 | 0 | 30K |
| ▶ PAG only | PAG | 3.0 | mid (m0) | – | – | – | – | 30K |
| ▶ SEG only | SEG | 3.0 | mid_block | – | – | – | – | 30K |
| ▶ SSG only | SSG | 1.0 | all attn | – | – | – | – | 30K |
| **Plug-and-Play** | | | | | | | | |
| ▶ PAG + TAG | PAG | 3.0 | mid (m0) | 1.15 | 1.0 | 750 | 450 | 30K |
| ▶ SEG + TAG | SEG | 3.0 | mid_block | 1.15 | 1.0 | 750 | 450 | 30K |
| ▶ SSG + TAG | SSG | 1.0 | all attn | 1.08 | 1.0 | 1000 | 750 | 30K |

**SEG + TAG.** Smoothed Energy Guidance (SEG) (Hong, 2024) applies Gaussian blur to the self-attention query projections to create a structurally degraded prediction. We combine SEG with TAG analogously: SEG guidance is applied at the noise-prediction level (`seg_scale= 3.0`, `blur_sigma= 100.0`, mid-block only), followed by TAG decomposition after the scheduler step.

**SSG + TAG.** Self-Swap Guidance (SSG) (Zhang et al., 2026) stochastically swaps a small fraction of token and channel positions in the self-attention activations to create a structurally perturbed prediction. We combine SSG with TAG analogously: SSG guidance is applied at the noise-prediction level with `ssg_scale= 1.0` and `swap_ratio= 0.001` across all 16 UNet self-attention layers (`d0 .. d5`, `m0`, `u0 .. u8`), followed by TAG decomposition after the scheduler step.

Table 17 summarizes the configurations for the PAG, SEG and SSG compatibility experiments. All experiments use unconditional SD v1.5 generation with 30,000 samples and the PNDM scheduler.

### F.5. Evaluation Metrics

**Fréchet Inception Distance (FID) and Inception Score (IS).** We compute FID and IS using `torch-fidelity` 0.4.0 (Obukhov et al., 2020) with the `inception-v3-compat` feature extractor (feature layer 2048 for FID, `logits_unbiased` for IS). All generated images are resized to $299 \times 299$ before feature extraction. For unconditional experiments, we use the full MS-COCO 2014 validation set (40,504 images) as the reference distribution. For conditional experiments, we use a 10,000-image subset of MS-COCO 2014 validation images aligned with the prompt set. Reference statistics are precomputed and cached as `.npz` files.

**Aesthetic Score (AES).** We compute the LAION aesthetic score using the ImageReward (Xu et al., 2023) library (`RM.load_score("Aesthetic")`), which employs a linear probe trained on top of CLIP ViT-L/14 features with the SAC+LOGOS+AVA dataset. Scores are computed per-image and averaged over the full set.

**CMMD.** We compute the CLIP Maximum Mean Discrepancy (CMMD) (Jayasumana et al., 2024) using the `clip-mmd` 0.0.2 library, which extracts CLIP features from both generated and reference image sets and computes an MMD-based distributional distance. The reference set is the MS-COCO 2014 validation set (40,504 images).

**No-Reference Image Quality Assessment (NR-IQA).** We additionally report two NR-IQA metrics computed using the `pyiqa` 0.1.15 library (Chen & Mo, 2022):

- **CLIP-IQA** (Wang et al., 2023): CLIP-based perceptual quality score (higher is better).

- **MUSIQ** (Ke et al., 2021): Multi-scale image quality transformer trained on KonIQ-10k (higher is better).

**CLIPScore.** For conditional experiments, we compute CLIPScore using the ImageReward CLIP model (`RM.load_score("CLIP")`), which wraps CLIP ViT-L/14. The raw cosine similarity between image and text embeddings is scaled by 100.

**ImageReward.** We use the ImageReward-v1.0 model (`RM.load("ImageReward-v1.0")`) (Xu et al., 2023), which is a BLIP-based reward model fine-tuned on human preference data. For conditional experiments, scores are computed per prompt–image pair and averaged.

### F.6. Software and Hardware

All generation and evaluation experiments are conducted on NVIDIA GeForce RTX 4090 GPUs (24 GB VRAM). Multiple GPUs are used in parallel for generation (data-parallel across seeds) but each individual image is generated on a single GPU without model parallelism.

| Package | Version |
|---|---|
| PyTorch (Paszke et al., 2019) | 2.5.1 (CUDA 12.1) |
| Diffusers (von Platen et al., 2022) | 0.35.1 |
| Transformers | 4.46.3 |
| Accelerate | 1.5.2 |
| torch-fidelity (Obukhov et al., 2020) | 0.4.0 |
| pyiqa (Chen & Mo, 2022) | 0.1.15 |
| clip-mmd | 0.0.2 |
| ImageReward (Xu et al., 2023) | 1.5 |

