# OpenReview forum: "TAG: Tangential Amplifying Guidance for Hallucination-Resistant Sampling"
_ICML.cc/2026/Conference — ICML 2026 regular_

### Official Review · Reviewer_mvkD · 2026-03-09

**Soundness:** 3
**Presentation:** 2
**Significance:** 3
**Originality:** 3
**Overall Recommendation:** 4
**Confidence:** 3

**Summary:**

This paper establishes a link between the score’s intrinsic geometry and sample quality, proving that amplifying the tangential components steers sampling trajectories toward the in-distribution manifold, which can guide the diffusion sampling to reduce hallucinations.

Based on the above analysis and motivation, the proposed method, Tangential Amplifying Guidance (TAG), is theoretically grounded, computationally lightweight, and architecture-agnostic.

**Compliance With Llm Reviewing Policy:**

Affirmed.

**Final Justification:**

All my concerns are solved.

**Key Questions For Authors:**

1. Performance summarization. The proposed method do not improve different metrics consistently on modern architectures (table 9), it is necessary to summarize the performance on different models in the abstract, contribution and conclusion part.

2. Baseline implementation. Why the FID reported in the manuscript so different from those reported in APG and TCFG? The FID on MS-COCO and ImageNet are apparently worse than those reported in APG and TCFG.

**Limitations:**

yes

**Strengths And Weaknesses:**

Strength
The proposed method is well motivated. The observation that the tangential component carries rich structural information (Figure 2), and amplifying it reduces out-of-distribution samples (Figure 3) is very interesting.

The theoretical analysis is insightful, establishing a solid framework for training free diffusion guidance from geometric view.



The proposed method is plug-and-play, and is evaluated with different diffusion/flow matching models including EDM2, Stable Diffusion v1.5/v2.1/XL, and Stable Diffusion 3.



Weakness
The proposed method do not improve different metrics consistently on modern architectures (table 9), it is necessary to summarize the performance on different models in the abstract, contribution and conclusion part.

Why the FID reported in the manuscript so different from those reported in APG and TCFG? The FID on MS-COCO and ImageNet are apparently worse than those reported in APG and TCFG.

---

> ### Author Rebuttal · Authors · 2026-03-31
>
> **Dear reviewer mvkD,**
>
> We thank the reviewer for the thoughtful feedback. We address the concerns below
>
> ---
>
> ## **C1. Performance Summarization**
>
> For clarity, we summarize the overall results in a consolidated table, which shows a broadly consistent improvement from TAG across settings. We also re-ran the Table 9 setting and include the updated result.
>
> ---
>
> # **Summary of TAG performance across different regimes**
>
> **(A) Uncond. Gen. (FID↓, No-Reference IQA↑) [KhKj.C1, C5Y8.C1]**
>
> |Setting|Model|Metric|w/o TAG|w/ TAG|Gain|Dataset|
> |-|-|-|-|-|-|-|
> |Uncond|SD1.5|FID (30K, COCO)|58.41|**46.20**|↓20.9%|COCO|
> |Uncond|SD2.1|FID (30K, COCO)|78.54|**59.94**|↓23.7%|COCO|
> |Uncond|SDXL|FID (30K, COCO)|119.14|**90.71**|↓23.9%|COCO|
> |Uncond|SD3|FID (30K, COCO)|84.26|**79.11**|↓6.1%|COCO|
> |Uncond|EDM2|FID (50K, ImageNet)|11.04|**10.37**|↓6.1%|ImageNet|
> |Uncond|Qwen-Image|CLIPIQA↑|0.49|**0.50**|+2.1%|COCO|
> |Uncond|Qwen-Image|MUSIQ↑|67.90|**68.85**|+1.4%|COCO|
>
> **Note:** CLIPIQA and MUSIQ are standard no-reference image quality metrics. We report them for Qwen-Image as FID on MS-COCO may not fully reflect perceptual quality for recent models due to distribution mismatch. See Reviewer C5Y8 (C1 & C3) for full metrics.
>
> **(B) Conditional Generation (Low CFG, FID ↓)**
>
> |Setting|Model|w/o TAG|w/ TAG|Gain (FID↓)|Dataset|
> |-|-|-|-|-|-|
> |Cond|SD1.5|26.2|**23.4**|↓10.6%|COCO |
> |Cond|SD2.1|28.24|**25.53**|↓9.6%|COCO |
> |Cond|SD3|29.02|**27.54**|↓5.1%|COCO|
>
> All results use a low CFG scale ($w=2.5$ for SD1.5 & SD2.1), following standard FID protocols. We separate conditional results by CFG regime, as low CFG favors metrics based on distribution similarity (e.g., FID), while higher CFG improves alignment and perceptual quality.
>
> **(C) Conditional Generation (Standard CFG, Alignment & Preference ↑) [SHmd.C23]**
>
> |Setting|Model|Metric|w/o TAG|w/ TAG|Gain(%)|Dataset|
> |-|-|-|-|-|-|-|
> |Cond|SDXL|CLIPScore|26.33|**26.64**|+1.2%|COCO|
> |Cond|SDXL|ImageReward|0.648|**0.743**|+14.7%|COCO|
> |Cond|SD3|CLIPScore|26.39|**26.56**|+0.6%|COCO |
> |Cond|SD3|ImageReward|1.030|**1.043**|+1.3%|COCO|
> |Cond|SDXL|Spatial Score|0.1857|**0.1980**|+6.6%|T2I-CompBench|
> |Cond|SDXL|BLIP-VQA|0.4443|**0.4650**|+4.7%|T2I-CompBench|
> |Cond|SDXL|3-in-1 Score|0.3364|**0.3472**|+3.2%|T2I-CompBench|
> |Cond|Z-Image|CLIPScore|**26.776**|26.763|-0.04%|COCO|
> |Cond|Z-Image|ImageReward|0.0323|**0.0344**|+6.5%|COCO|
>
> All results use a CFG scale of $w=5$, a standard setting for alignment-focused generation, rather than the lower CFG used for FID evaluation.
>
> **(D) Video Generation Results [C5Y8.C1]**
>
> |Setting|Model|Metric|w/o TAG|w/ TAG|Gain (%)|Dataset|
> |-|-|-|-|-|-|-|
> |Video|Wan2.2|Imaging Quality ↑|0.7047|**0.7066**|+0.26%|VBench|
> |Video|Wan2.2|Aesthetic Quality ↑|0.5647|**0.5697**|+0.8%|VBench|
> |Video|Wan2.2|Overall Consistency ↑|0.1796|**0.1836**|**+2.2%**|VBench|
> |Video|Wan2.2|Dynamic Degree ↑|0.52|**0.56**|**+7.6%**|VBench|
>
> TAG improves not only image metrics but also **video dynamics** and **consistency**, indicating that tangential amplification *generalizes beyond static image generation.*
>
> ---
>
> **Clarification on Table 9.** Table 9 uses a high CFG setting ($w=7.5$), where stronger guidance improves perceptual quality but often degrades FID—a well-known trade-off in prior work (e.g., ADM, PAG, SEG).
>
> To ensure a fair comparison under standard FID protocols, we re-evaluate using a lower CFG scale ($w=2.5$), consistent with Table 3. Under this setting, TAG consistently improves FID:
>
> **FID@10K results on SD 2.1 ($w=2.5,\eta=1.4$)**:
>
> ||w/o TAG|w/ TAG|Improvement|
> |-|-|-|-|
> |SD2.1|28.24|**25.53**|**9.6%**|
>
> This confirms that the apparent inconsistency in Table 9 arises from the high-CFG evaluation setting, rather than a limitation of TAG. Under standard evaluation, TAG shows consistent improvements across models, in line with Table 1 and Table 3.
>
> We will revise the abstract and conclusion to clearly summarize this behavior across models and settings.
>
> ---
> ## **C2. On differences in reported FID**
>
> The discrepancy mainly stems from differences in evaluation settings.
>
> For ImageNet results (Table 1), we evaluate TAG under an *unconditional* sampling setting using Stable Diffusion backbones, rather than fully optimized ImageNet-trained conditional models (e.g., ADM/EDM), which typically report FID < 10. This setup explains the gap.
>
> In contrast, prior works such as APG and TCFG evaluate SD-family models on MS-COCO under conditional generation. We additionally evaluate on MS-COCO, which better matches the training distribution of SD models, and observe consistent improvements across all models, as summarized in **C1 (A) Unconditional Generation.**
>
> Our baseline FIDs are consistent with prior work under comparable settings (e.g., PAG: 53.13 on SD1.5; SEG: 129.50 on SDXL). Remaining differences with APG/TCFG likely stem from protocol variations, including dataset splits, evaluation size, and mainly conditional vs. unconditional settings.

---

> > ### Author Rebuttal · Reviewer_mvkD · 2026-04-02
> >
> > All my concerns are solved.

---

> > > ### Author Response · Authors · 2026-04-02
> > >
> > > Dear Reviewer mvkD,
> > >
> > > We sincerely appreciate your thoughtful update and your careful reconsideration of our rebuttal. We are especially pleased to see that your assessment has been revised from weak reject(3) to **weak accept(4)**, and **we are very glad that all of your concerns were resolved** through the discussion.
> > >
> > > We also greatly appreciate your time, constructive feedback, and generous evaluation throughout the review process.
> > >
> > > Sincerely,
> > > The Authors

---

### Official Review · Reviewer_SHmd · 2026-03-09

**Soundness:** 3
**Presentation:** 3
**Significance:** 3
**Originality:** 3
**Overall Recommendation:** 4
**Confidence:** 3

**Summary:**

This paper proposes Tangential Amplifying Guidance (TAG), an inference-time modification to diffusion sampling that amplifies the tangential component of the solver update with respect to an iso-noise manifold. The method decomposes each sampling update into normal and tangential components and increases the weight of the tangential direction to steer trajectories toward higher-density regions of the data manifold.

**Compliance With Llm Reviewing Policy:**

Affirmed.

**Final Justification:**

Thanks for the detailed rebuttal, which addressed my primary concerns regarding the sensitivity of the amplification factor and the performance on targeted hallucination benchmarks. So I am maintaining my score of Weak Accept.

**Key Questions For Authors:**

1. Can the amplification factor η be adapted automatically during sampling?

2. How does TAG perform on benchmarks designed for compositional reasoning or hallucination detection (such as T2I-CompBench)?

3. The ablation study in Table 4 is limited to unconditional generation using SD1.5. It is recommended to add an ablation study on η under text-conditional generation using SDXL/SD3.

**Limitations:**

The robustness of the method's performance requires further validation. Additionally, the empirical evaluation could be strengthened with more targeted hallucination benchmarks.

**Strengths And Weaknesses:**

Strengths

1.	TAG is easy to integrate into existing diffusion samplers and requires only lightweight vector projections without additional network evaluations.

2.	The paper provides an intuitive geometric perspective on diffusion sampling by decomposing updates into radial and tangential components.

Weaknesses

1.	The performance of TAG depends on the amplification factor η. Large values can destabilize the sampling trajectory and degrade sample quality.

2.	While the paper heavily emphasizes "Hallucination-Resistant" generation, the experiments rely entirely on general metrics (FID, IS, CLIPScore, and ImageReward) that cannot precisely measure structural hallucinations, alignment.

3.	The theoretical analysis relies on first-order approximations and assumes accurate score estimation. It remains unclear whether the claimed benefits hold under realistic score estimation errors.

---

> ### Author Rebuttal · Authors · 2026-03-31
>
> **Dear reviewer SHmd,**
>
> We thank the reviewer for the thoughtful feedback. We address the concerns below
>
> ---
> ## **C1. Sensitivity of $\eta$, practical selection guidelines, and adaptive $\eta$ scaling**
>
> **Understanding and mitigating sensitivity.** Selecting $\eta$ is analogous to CFG: increasing $\eta$ transitions from weak guidance to over-smoothing, with a suitable value in between.
>
> This can be understood from the tangential update. As shown in Fig.2, it varies across timesteps: at mid-noise it adds meaningful structure, at high-noise it introduces coarse, blurry patterns, and at low-noise it becomes increasingly noise-like.
>
> We observe that $\eta$ sensitivity (e.g., over-smoothing in Fig.7) is mainly driven by amplifying the high-noise component. When $\eta$ is large, these updates are over-amplified, degrading fine structure. Excluding this regime (roughly $t\in[1000,800]$ in Fig. 2) and applying TAG only at mid-noise substantially mitigates this effect.
>
> Low-noise updates are similarly less informative. Focusing on mid-noise timesteps, while excluding the high-noise regime, yields more stable behavior across a wide range of $\eta$, as shown below (SD 2.1, conditional generation, FID@10K on MS-COCO):
>
> |$\eta$|Fixed$\eta$ (full)|Fixed$\eta$ [800, 400]|Adaptive$\eta$ (full)|
> |-|-|-|-|
> |0.0|28.24|28.24|28.24|
> |0.5|26.30|26.64|26.91|
> |1.0|**25.39**|25.77|26.34|
> |1.5|25.85|25.29|26.05|
> |2.0|26.61|25.10|25.90|
> |2.5|27.82|**25.08**|**25.65**|
> |3.0|29.63|25.17|25.86|
>
> **Adaptive selection of** $\eta$ **during sampling.**
> While $\eta$ sensitivity is already mitigated by mid-noise windowing, we further explore an adaptive formulation.
>
> In the C-TAG update, $\tilde{\epsilon}_k = \epsilon_u + \omega g_k + \eta g_k^{\mathrm{align}}$, where $g_k$ is the CFG guidance term and $g_k^{\mathrm{align}}$ the tangential component. Since $\|g_k^{\mathrm{align}}\|$ varies across timesteps, a fixed $\eta$ yields inconsistent update magnitudes.
>
> We rescale $g_k^{\mathrm{align}}$ by $\|g_k\|/\|g_k^{\mathrm{align}}\|$, so its magnitude follows $\|g_k\|$. This aligns the TAG correction with the guidance scale, while $\eta$ controls only relative amplification.
>
> Empirically, this reduces sensitivity to $\eta$ and yields stable behavior over a wider range, achieving comparable robustness without explicit guidance window selection (see the Adaptive column above).
>
> ---
> ## **C2. TAG on hallucination detection benchmark**
>
> > **For a comprehensive summary of the overall results, please refer to the consolidated table (mvkD).**
>
> We evaluate TAG on T2I-CompBench, which measures compositional faithfulness in T2I generation. We focus on **Spatial** and **Complex** subsets, where structural hallucinations are most common (e.g., object placement and multi-object relations).
>
> - AES: Aesthetic
> - CLIP: CLIPScore
> - IR: ImageReward
>
> |Spatial-Val300|2DSpatial↑|AES↑|CLIP↑|IR↑|
> |-|-|-|-|-|
> |SDXL|0.1857|**5.779**|27.365|0.800|
> |+TAG($\eta=0.3$)|**0.1980(+6.6%)**|5.768(-0.2%)|**27.714(+1.3%)**|**0.911(+13.9%)**|
>
> |Complex-Val300|BLIP-VQA↑|2DSpatial↑|ComplexCLIP↑|3-in-1↑|AES↑|CLIP↑|IR↑|
> |-|-|-|-|-|-|-|-|
> |SDXL|0.4443|**0.0243**|0.2910|0.3364|5.666|25.975|0.2596|
> |+TAG($\eta=0.5$)|**0.4650(+4.7%)**|0.0232(-4.5%)|**0.2937(+0.9%)**|**0.3472(+3.2%)**|**5.667(+0.0%)**|**26.477(+1.9%)**|**0.3978(+53.2%)**|
>
> TAG improves compositional metrics in both subsets: spatial score, CLIP, and IR on Spatial, and BLIP-VQA, ComplexCLIP, and 3-in-1 on Complex.
>
> Consistent with these gains, we observe that TAG suppresses common structural artifacts in hallucination-prone prompts (e.g., texture-like scattered objects), leading to more coherent object structure. This indicates improved compositional faithfulness rather than merely altering perceptual style.
>
> ---
> ## **C3. $\eta$ sweep for T2I generation**
>
> We ablate $\eta$ on a 1K MS-COCO subset. For both **SDXL** and **SD3**, moderate $\eta$ improves CLIP and IR over the baseline.
>
> **SDXL results:**
>
> |Metric / $\eta$|baseline|0.5|1.0|1.75|2.0|3.5|
> |-|-|-|-|-|-|-|
> |AES↑|5.598|5.590 (-0.1%)|5.595 (-0.1%)|5.597 (-0.0%)|5.604 (+0.1%)|5.615(+0.3%)|
> |CLIP↑|26.326|26.472 (+0.6%)|26.602 (+1.0%)|26.644 (+1.2%)|26.593 (+1.0%)|26.422 (+0.4%)|
> |IR↑|0.648|0.683 (+5.4%)|0.716 (+10.5%)|0.739 (+14.0%)|0.743 (+14.7%)|0.694 (+7.1%)|
>
> **SD3 results:**
>
> ||baseline|0.5|0.75|1.0|1.5|
> |-|-|-|-|-|-|
> |AES↑|5.403|5.434 (+0.6%)|5.442 (+0.7%)|5.451 (+0.9%)|5.461 (+1.1%)|
> |CLIP ↑|26.386|26.450 (+0.2%)|26.548 (+0.6%)|26.557 (+0.6%)|26.504 (+0.4%)|
> |IR↑|1.030|1.042 (+1.2%)|1.043 (+1.3%)|1.043 (+1.3%)|1.024(-0.6%)|
>
> Both models perform best at moderate $\eta$, with weaker gains at larger values, indicating a stable operating range.
>
> We further report SD3 FID on 10K MS-COCO prompts using the window [800,400] (Sec. C1). Under this setting, increasing $\eta$ consistently improves FID.
>
> |Metric|baseline|0.5|1.0|1.5|2.0|2.5|3.0|3.5|4.0|4.5|5.0|
> |-|-|-|-|-|-|-|-|-|-|-|-|
> |FID↓|29.02|28.77|28.40|28.12|27.95|27.69|27.54|27.34|27.26|27.15|**27.05(+6.8%)**|

---

> > ### Author Rebuttal · Reviewer_SHmd · 2026-04-03
> >
> > Thanks for the detailed rebuttal, which addressed my primary concerns regarding the sensitivity of the amplification factor and the performance on targeted hallucination benchmarks. So I am maintaining my score of Weak Accept.

---

> > > ### Author Response · Authors · 2026-04-08
> > >
> > > Dear Reviewer SHmd,
> > >
> > > Thank you for your positive assessment of our rebuttal. We are pleased that our clarifications helped address your primary concerns, particularly regarding the sensitivity of the amplification factor and the performance on targeted hallucination benchmarks.
> > >
> > > We will make sure these clarifications and results are clearly reflected in the final version of the paper.
> > >
> > > Sincerely,
> > > The Authors

---

### Official Review · Reviewer_C5Y8 · 2026-03-13

**Soundness:** 3
**Presentation:** 3
**Significance:** 2
**Originality:** 3
**Overall Recommendation:** 3
**Confidence:** 3

**Summary:**

The paper introduces Tangential Amplifying Guidance (TAG), a training-free module designed to reduce hallucinations in diffusion sampling. The method is theoretically well-motivated, leveraging a geometric decomposition to amplify semantic-rich tangential components. While the theoretical framework is solid and generalizable, the empirical evaluation significantly undermines the paper's claims due to outdated baselines.

**Compliance With Llm Reviewing Policy:**

Affirmed.

**Final Justification:**

I am skeptical about the effectiveness of this method on larger models.

**Key Questions For Authors:**

(1) Given the reliance on aging benchmarks like SD3 and EDM, can the authors demonstrate TAG’s effectiveness on latest 2025/2026 architectures where structural hallucinations are already significantly mitigated?
(2) Could the authors explain the unusually high baseline FID scores (e.g., 100+ vs. the standard <10), and how this affects the reliability of the claimed improvements?
(3) Why is a trajectory-based guidance like TAG still necessary for contemporary unified models that rarely exhibit the simple semantic errors shown in your qualitative examples?
(4) Is there a robust mechanism or theoretical principle to determine the optimal amplification factor $\eta$ beyond extensive empirical tuning?

**Limitations:**

Please refer to the weakness.

**Strengths And Weaknesses:**

### strength
(1) The paper is well-organized and provides a rigorous geometric interpretation of the sampling trajectory, which is potentially applicable beyond the image domain.
(2) As a training-free method, TAG offers high utility for immediate integration into existing inference pipelines.
(3) TAG only involves simple vector projections. This makes it exceptionally efficient for real-time applications as it introduces negligible latency to the sampling process.
### weakness
(1) The use of ImageNet-1k and MS-COCO (2014) is insufficient for a 2026 submission. Evaluating primarily on older architectures like EDM and SD3 ignores the rapid progress in 2024-2026 (e.g., Qwen-Image, Z-Image). Demonstrating gains on weak or aging baselines offers limited practical value for the current state-of-the-art.
(2) The reported FID scores are unusually high compared to standard literature for the cited models. This discrepancy requires clarification, as it raises concerns about the reliability of the evaluation pipeline.
(3) The "hallucinations" shown in the examples are often absent in recent SOTA T2I models. To prove TAG's necessity, the authors should demonstrate that it resolves failure cases in more advanced, contemporary architectures.
(4) The effectiveness of TAG heavily relies on the amplification factor $\eta$. The manuscript lacks a systematic guide or an adaptive mechanism for parameter selection.

---

> ### Author Rebuttal · Authors · 2026-03-31
>
> Dear reviewer C5Y8,
>
> We thank the reviewer for the thoughtful feedback. We address the concerns below
>
> ---
>
> ## **C1,3. TAG on cutting-edge models (Image & Video)**
>
> > **Please refer to the consolidated table (mvkD) for a comprehensive summary of the overall results.**
>
> We fully agree that recent T2I diffusion models (e.g., Qwen-Image, Z-Image) have made remarkable progress in reducing artifacts such as extra fingers or structural inconsistencies, and we appreciate these advances.
>
> Recent works, including Z-Image, have clearly demonstrated the importance of data curation and additional SFT stages with highly curated datasets in achieving these improvements[1].
>
> Our work is complementary and orthogonal to this line of research. Rather than relying on improved training data or additional fine-tuning stages, TAG operates purely at inference time and provides a principled, geometry-based approach to reducing hallucinations. This allows it to improve sample quality without modifying the training pipeline.
>
> Following the reviewer’s suggestion, we evaluate TAG on a recent SOTA model (Qwen-Image). As shown below, TAG consistently improves performance even on this strong backbone, demonstrating that its effectiveness extends to contemporary models.
>
> |Uncond. 10K|CLIPIQA↑|MUSIQ↑|FID↓|
> |-|-|-|-|
> |Qwen-I|0.4947|67.90|**57.53**|
> |+TAG|**0.5051(+2.1%)**|**68.85(+1.4%)**|59.84(+4.0%)|
>
> Here, we use $\eta=1.15$ over timesteps $[1000,600]$. While FID slightly degrades in this setting, we consistently observe improved structural coherence (e.g., correcting common structural errors such as distorted anatomy and faces) and reduced artifacts across diverse samples.
>
> We note that FID computed on MS-COCO may not fully reflect these improvements, as the training distribution of Qwen-Image likely differs substantially from MS-COCO. To better capture perceptual and structural quality, we additionally report no-reference image quality assessment metrics, including CLIP-IQA[3] and MUSIQ[4], both of which show consistent improvements.
>
> More broadly, as the reviewer notes, TAG’s geometric perspective may extend beyond images, offering a complementary path to data-centric methods through principled trajectory control across models and modalities.
>
>
> **Evaluation on a cutting-edge video model.** To further validate this generality, we evaluated TAG on Wan2.2 using 100 randomly sampled VBench prompts:
>
> ||window|Dynamic Degree↑|Imaging Quality↑|Aesthetic Quality↑|Background Consistency↑|Subject Consistency↑|Human Action ↑|Motion Smoothness↑|Overall Consistency↑|Temporal Flickering↑|Temporal Style↑|
> |-|-|-|-|-|-|-|-|-|-|-|-|
> |Wan2.2||0.52|0.7047|0.5647|0.9644|0.9613|0.05|**0.9871**|0.1796|**0.9772**|0.1796|
> |+TAG (1.1)|1000,400|0.54|**0.7092**|0.5687|0.9660|**0.9632**|0.05|0.9869|0.1819|0.9767|0.1819|
> |+TAG (1.2)|1000,400|**0.56**|0.7066|**0.5697**|**0.9672**|0.9625|**0.07**|0.9864|**0.1836**|0.9759|**0.1836**|
>
> Both TAG variants improve over the baseline on most metrics, including dynamic degree, imaging quality, aesthetic quality, consistency, and temporal style. These results suggest that TAG supports more dynamic video generation without simply compressing motion to achieve better quality and consistency — as evidenced by the increased dynamic degree alongside the quality gains.
>
> From a qualitative perspective, we observe that Wan2.2 occasionally produces structural artifacts in scenes involving complex object compositions (e.g., a person riding a bicycle with a vehicle in the background), whereas TAG reliably preserves the coherence of such scenes.
>
> We will include qualitative results as well as the additional metrics and discussion in the final version.
>
> ---
> ## **C2. Clarification of FID**
>
> Due to space constraints, we refer the reviewer to **our response to Reviewer KhKj (C1)** for a more detailed discussion. Here, we briefly summarize the key points:
>
> 1. Very low FID scores (e.g., <10) are typically reported for ImageNet-trained models, often under conditional generation settings, where the evaluation distribution closely matches the training data, resulting in minimal distribution mismatch.
>
> 2. We re-evaluated our text-to-image models under a more aligned evaluation setup (MS-COCO), and the resulting FID values are consistent with reported FIDs in prior work.
>
> 3. Under this aligned setting, TAG yields consistent improvements across all models.
>
> ---
> ## **C4. Selection of $\eta$**
>
> Due to space constraints, we refer to **Reviewer SHmd (C1)** for details. Here we summarize:
>
> 1. $\eta$ is simple and predictable to tune, similar to CFG.
>
> 2. Sensitivity mainly arises from high-noise timesteps; restricting TAG to mid-noise mitigates over-smoothing and enables larger $\eta$.
>
> 3. We also explore a preliminary adaptive $\eta$ scheme that improves robustness without manual tuning.
>
> ---
>
> [1] https://www.krea.ai/blog/flux-krea-open-source-release
>
> [2] arXiv.2511.22699
>
> [3] arXiv.2207.12396
>
> [4] arXiv.2108.05997

---

> > ### Author Rebuttal · Reviewer_C5Y8 · 2026-04-08
> >
> > I am skeptical about the effectiveness of this method on larger models.

---

> > > ### Author Response · Authors · 2026-04-08
> > >
> > > Dear Reviewer C5Y8,
> > >
> > > Qualitative results on Video generation :https://anonenini.github.io/anone/
> > >
> > > Thank you for your continued engagement with our work. We sincerely appreciate your time and effort throughout the review process.
> > >
> > > First, we acknowledge your concern regarding the effectiveness of TAG on larger, modern models. To provide a balanced assessment, we summarize both the observed improvements and the current limitations below.
> > >
> > > ### **Current limitations on modern models**
> > >
> > > 1. **Qwen-Image**: FID worsens slightly by **4.0%**, which may be due to a distribution mismatch between the model’s training data and MS-COCO.
> > >
> > > ### **What TAG improves on modern models**
> > >
> > > 1. **Qwen-Image**: CLIP-IQA improves by **2.1%**, and MUSIQ improves by **1.4%**.
> > > 2. **Wan2.2 (video)**: TAG improves imaging quality, aesthetic quality, dynamic degree by **7.6%**, and overall consistency by **2.2%**.
> > > 3. **Z-Image (reviewer mvkD)**: ImageReward improves by **6.5%**.
> > >
> > > ### **Overall take-away**
> > >
> > > We agree that the gains on larger models are more modest than those on earlier architectures.
> > >
> > > At the same time, we believe the consistent improvements in perceptual quality metrics (e.g., CLIP-IQA, MUSIQ, and ImageReward), together with the extension to video generation, indicate that TAG still **provides useful benefits** even on strong contemporary backbones.
> > >
> > > Sincerely,
> > > The Authors

---

### Official Review · Reviewer_KhKj · 2026-03-23

**Soundness:** 3
**Presentation:** 3
**Significance:** 3
**Originality:** 3
**Overall Recommendation:** 4
**Confidence:** 4

**Summary:**

The authors propose Tangential Amplifying Guidance (TAG), a training free inference method which can be applied for conditional and unconditional diffusion sampling. The goal is to decompose the update step in a diffusion sampler into tangential and radial components and to amplify the tangential component which encodes the semantic structure of the image thus reducing hallucinations or generative artifacts.

**Compliance With Llm Reviewing Policy:**

Affirmed.

**Final Justification:**

Based on the authors response, most of my concerns seem to be addressed

**Key Questions For Authors:**

See weaknesses

**Limitations:**

yes

**Strengths And Weaknesses:**

# Strengths
- **[Presentation]:**
The presentation is quite clear in terms of the theoretical and intuitive arguments (using figures etc.)
- **[Significance]:**
The paper tackles an important problem of reducing hallucinations or atifacts in diffusion model sampling.
- **[Originality]:**
To the best of my knowledge, the presented theoretical arguments in the paper are novel and original.

# Weaknesses

**TLDR**. I think the paper presents a good idea supported with good theoretical arguments. Though, I have some concerns regarding the empirical results of the method due to which I would recommend a weak reject.

- **[Soundness Issues]**
    - **Empirical Results**:
        - Firstly, in Table 1 why are the FID scores on the ImageNet validation set are so high? Usually in state of the art generative models the FID scores for ImageNet are <5. Since the technique presented in this paper is training free can the authors apply it to a state of the art diffusion model trained on ImageNet (maybe EDM-2?) and report results using that in Table 1. Since the upstream diffusion model is so bad at this task, its hard to assess if the method actually provides any meaningful improvements. I think its fine if the improvements are modest and I feel like the current empirical results seem to exaggerate the severity of the problem. Moreover, while the authors do not report results using EDM-2 in Table 1, they do report qualitative results using this model in Fig. 11 in the Appendix. Why is that the case?

       - Secondly, for unconditional sampling, the proposed method is analogous to a corrector scheme in diffusion sampling. There is a large body of work which aims to tackle the problem of inaccurate unconditional sampling in diffusion models like PC sampling (Song et al.). However, the authors have not compared the proposed method with any of these baselines in the paper.

    - **Sampling guarantees** - Are there any theoretical results in the paper which suggest that the sampling process is guaranteed to sample from the underlying data distribution p(x) for unconditional sampling and p(x|y) for conditional sampling? Moreover, it would be great if the authors could comment on how to tune the parameter $\eta$ in practice as from Figure 7, it seems like the method is quite sensitive to small perturbations in $\eta$? In other words, what is the impact of $\eta$ on the convergence rates of the sampling process? Are there any formal guarantees?

---

> ### Author Rebuttal · Authors · 2026-03-31
>
> Dear reviewer KhKj,
>
> We thank the reviewer for the thoughtful feedback. We address the concerns below.
>
> ---
>
> ## **C1. Clarification of FID & EDM2 Validation**
>
> > **For a comprehensive summary of the overall results, please refer to the consolidated table (mvkD).**
>
> **FID in Table 1.** We clarify that Table 1 evaluates TAG under an *unconditional sampling setting* using Stable Diffusion backbones, rather than fully optimized *ImageNet-trained conditional* generative models (e.g., ADM/EDM), which typically report FID < 10.
>
> That said, we agree that evaluating on datasets aligned with the training distribution of SD models provides stronger insight. Following the reviewer’s suggestion, we additionally compute FID on MS-COCO, where SD-family models are more naturally evaluated. We observe consistent and significant improvements across all models:
>
> |Uncond. 30K|$\eta$|w/o TAG|w/ TAG|Improvement|
> |-|-|-|-|-|
> |SD1.5|1.15|58.41|**46.20**|**20.9%**|
> |SD2.1|1.15|78.54|**59.94**|**23.7%**|
> |SDXL|1.20|119.14|**90.71**|**23.9%**|
> |SD3|1.05|84.26|**79.11**|**6.1%**|
>
> *For reference, the baseline FID values are **broadly consistent** with prior guidance results reported under comparable settings. e.g., PAG reports FID@30K = 53.13 on SD1.5, and SEG reports FID@30K = 129.50 on SDXL.*
>
> These results further support that TAG consistently improves sample quality across diverse backbones. The relatively modest gain on SD3 is likely attributable to differences in training distribution and objectives, including improvements in text rendering (e.g., typography), which can affect alignment with MS-COCO evaluation.
>
> In addition to FID, we also report CMMD (Jayasumana et al.), a CLIP-based distributional metric, and Aesthetic score, a learned predictor of visual appeal, to complement fidelity with perceptual similarity and human-preference-oriented evaluation. As shown below and Figs.6/11, TAG consistently reduces artifacts and improves perceptual quality. We will include these results and clarify the evaluation protocol in the final version.
>
> ||AES↑|CMMD↓|
> |-|-|-|
> |SD3|5.261|1.671|
> |+ TAG|**5.365**|**1.564**|
>
> **Results on a strong ImageNet-trained model (EDM2).** We observe consistent improvement in FID without additional NFEs:
>
> |Uncond. 50K|$[\eta_{\rm sta},\eta_{\rm end}]$|scale $\eta$|FID↓|
> |-|-|-|-|
> |EDM2|–|–|11.0432|
> |+TAG|[500, 250]|1.3|**10.3741**|
>
> ---
>
> ## **C2. Comparison to other sampling methods**
>
> To better position TAG with respect to existing improvements for unconditional diffusion sampling, we additionally compared it against representative sampler variants and also evaluated their combination with TAG. All experiments are conducted with $\eta=1.15$.
>
> We first compare TAG with stronger sampling methods. TAG alone improves substantially over the baseline and is competitive with widely used sampler improvements such as DPM++ (order 3) and UniPC:
>
> |SD1.4|FID@30K|IS|
> |-|-|-|
> |Baseline|65.55|14.53±0.32|
> |*TAG*|52.83|16.16±0.31|
> |*DPM++*|55.40|15.92±0.25|
> |*UniPC*|50.82|16.37±0.25|
>
> We then apply TAG on top of these stronger samplers. In both cases, TAG provides further gains:
>
> |SD1.4|FID@30K|IS|
> |-|-|-|
> |UniPC *+TAG*|48.34|16.86±0.36|
> |DPM++ *+TAG* |44.08|17.77±0.36|
>
> These results place TAG as a lightweight update that is competitive with stronger sampler variants and also complementary to them. We will include these comparisons and clarify this positioning in the revision.
>
> ---
> ## **C3. Sampling guarantee**
>
> In score-based diffusion, exact sampling from $p(x)$ (or $p(x\mid y)$) is guaranteed only in the ideal setting of exact scores and exact reverse-time dynamics. With a finite-capacity pretrained model and test-time guidance, the goal is no longer exact distribution preservation, but steering trajectories away from low-density, hallucination-prone regions and toward reliable manifolds. This is also the lens taken by prior guidance work: CFG is understood as trading mode coverage for sample fidelity, and Karras et al. argue that limited-capacity score models can spread mass into low-probability outskirts, leading to outliers.
>
> In this sense, our theory shows that TAG biases trajectories toward higher-density regions, rather than proving exact recovery of $p(x)$. Figure 3 reflects this objective directly (also Fig.1 in Karras et al)=> TAG suppresses off-manifold outliers while preserving fine structure, and is closest to the GT branching distribution among the compared methods.
>
> ---
> ## **C4. Sensitivity and Adaptive Scaling of $\eta$**
> Due to space constraints, we refer to **Reviewer SHmd (C1)** for details. Here we summarize:
>
> (1) $\eta$ is simple and predictable to tune, similar to CFG.
>
> (2) Sensitivity mainly arises from high-noise timesteps; restricting TAG to mid-noise mitigates over-smoothing and enables larger $\eta$.
>
> (3) We also explore a preliminary adaptive $\eta$ scheme that improves robustness without manual tuning.

---

> > ### Author Rebuttal · Reviewer_KhKj · 2026-04-04
> >
> > Thank you for your response which addresses most of my concerns. I hope the authors can include additional evaluations using models actually trained on ImageNet

---

> > > ### Author Response · Authors · 2026-04-08
> > >
> > > Dear Reviewer KhKj,
> > >
> > > We are glad that our rebuttal addressed your concerns, and we sincerely thank you for updating your recommendation. Your thoughtful feedback throughout the review process has greatly strengthened our work.
> > >
> > > We appreciate your suggestion regarding broader ImageNet-native evaluation and will incorporate additional results where feasible in the final version.
> > >
> > > Sincerely,
> > > The Authors

---

### Decision · Program_Chairs · 2026-04-30

**Decision:**

Accept (regular)

**Comment:**

This paper presents TAG, a training-free geometric guidance method that amplifies tangential components of diffusion sampling updates to reduce hallucinations, backed by a rigorous theoretical framework that reviewers found novel and well-motivated. Main concerns included outdated baselines, high FID scores from evaluation mismatches, η sensitivity, and effectiveness on modern large-scale models. The authors provided extensive rebuttal experiments on contemporary models (Qwen-Image, Z-Image, Wan2.2), compositional benchmarks (T2I-CompBench), and an adaptive η scheme, well resolving three reviewers' concerns (with mvkD raising their score to Weak Accept). One reviewer remained unconvinced about TAG's necessity for state-of-the-art models, though the rebuttal evidence of consistent gains on Qwen-Image, Z-Image, and Wan2.2 was not specifically rebutted.
Considering the solid theoretical contribution, demonstrated generality across architectures and modalities, and the thorough rebuttal satisfying the clear majority of reviewers, I recommend acceptance.